# Determination of the multiple-scattering correction factor and its cross-sensitivity to scattering and wavelength dependence for different AE33 Aethalometer filter tapes: A multi-instrumental approach

Jesús Yus-Díez[1,2], Vera Bernardoni[3], Griša Močnik[4,5], Andrés Alastuey[1], Davide Ciniglia[3], Matic Ivančič[6], Xavier Querol[1], Noemí Perez[1], Cristina Reche[1], Martin Rigler[6], Roberta Vecchi[3], Sara Valentini[3], and Marco Pandolfi[1]

[1]Institute of Environmental Assessment and Water Research (IDAEA-CSIC), C/Jordi Girona 18-26, 08034, Barcelona, Spain
[2]Grup de Meteorologia, Departament de Física Aplicada, Universitat de Barcelona, C/Martí i Franquès, 1, 08028, Barcelona, Spain
[3]Dipartimento di Fisica "A. Pontremoli", Università degli Studi di Milano & INFN-Milan, via Celoria 16, 20133 Milano, Italy
[4]Center for Atmospheric Research, University of Nova Gorica, Vipavska 11c, SI-5270 Ajdovščina, Slovenia.
[5]Department of Condensed Matter Physics, Jozef Stefan Institute, Jamova 39, SI-1000 Ljubljana, Slovenia
[6]Aerosol d.o.o., Ljubljana, Slovenia

**Correspondence:** jesus.yus@idaea.csic.es

**Abstract.**

Providing reliable observations of aerosol particles absorption properties at spatial and temporal resolutions suited to climate models is of utter importance to better understand the effects that atmospheric particles have on climate. Nowadays, one of the instruments most widely used in international monitoring networks for in-situ surface measurements of light absorption properties of atmospheric aerosol particles is the multi-wavelength dual-spot aethalometer, AE33. The AE33 derives the absorption coefficients of aerosol particles at 7 different wavelengths from the measurements of the optical attenuation of light through a filter where particles are continuously collected. An accurate determination of the absorption coefficients from AE33 instrument relies on the quantification of the non-linear processes related to the sample collection on the filter. The multiple-scattering correction factor (C), which depends on the filter tape used and on the optical properties of the collected particles, is the parameter with both the greatest uncertainty and the greatest impact on the absorption coefficients derived from the AE33 measurements.

Here we present an in-depth analysis of the AE33 multiple-scattering correction factor C and its wavelength dependence for two different and widely used filter tapes, namely: the old, and most referenced, TFE-coated glass, or M8020, filter tape and the currently, and most widely used, M8060 filter tape. For performing this analysis, we compared the attenuation measurements from AE33 with the absorption coefficients measured with different filter-based techniques. Online co-located multi-angle absorption photometer (MAAP) measurements and offline PP_UniMI polar photometer measurements were employed as reference absorption measurements for this work. To this aim, we used data from three different measurement stations located in North-East of Spain, namely: an urban background station (Barcelona; BCN), a regional background station (Montseny;

MSY) and a mountain-top station (Montsec d'Ares; MSA). The median C values (at 637 nm) measured at the three stations

ranged between 2.29 (at BCN and MSY; lowest 5th percentile of 1.97 and highest 95th percentile of 2.68) and 2.51 (at MSA; lowest 5th percentile of 2.06 and highest 95th percentile of 3.06). The analysis of the cross-sensitivity to scattering, for the two filter tapes considered here, revealed a large increase of the C factor when the single scattering albedo (SSA) of the collected particles was above a given threshold, up to a 3-fold increase above the average C values. The SSA threshold appeared to be site-dependent, and ranged between 0.90 to 0.95 for the stations considered in the study. The results of the cross-sensitivity to

scattering displayed a fitted constant multiple scattering parameter, $C_f$, of 2.21 and 1.96, and a cross-sensitivity factor, $m_s$, of 1.8% and 3.4% for MSY and MSA stations, respectively, for the TFE-coated glass filter tape. For the M8060 filter tape, $C_f$ of 2.50, 1.96, 1.82 and $m_s$ of 1.6%, 3.0%, 4.9%, for BCN, MSY and MSA stations, respectively, were obtained. SSA variations also influenced the spectral dependence of the C, which showed an increase with wavelength when SSA was above the site-dependent threshold. Below the SSA threshold, no statistically significant dependence of the C with wavelength was observed.

For the measurement stations considered here, the wavelength-dependence of C was to some extent driven by the presence of dust particles during Saharan dust outbreaks that had the potential to increase the SSA above the average values. At the mountain-top station, an omission of the wavelength dependence of the C factor led to an underestimation of the Absorption Ångström Exponent (AAE) up to a 12%. Differences in the absorption coefficient determined from AE33 measurements at BCN, MSY and MSA of around a 35-40 % can be expected when using the site-dependent C experimentally obtained instead

of the nominal C value. Due to the fundamental role that the SSA of the particles collected on the filter tape has on the multiple scattering parameter C, we present a methodology that allows to recognize the conditions upon which the use of a constant and wavelength independent C is feasible.

# 1 Introduction

Atmospheric aerosol particles play an important role on the Earth's radiative balance directly by scattering and absorbing solar

and terrestrial radiation and indirectly by acting as cloud condensation nuclei. Large uncertainties still exist on the effects that atmospheric particles have on climate (Myhre et al., 2013). In fact, the aerosol-radiation interaction depends on aerosol properties such as aerosol size distribution, mixing state, and refractive index, among others (e.g. Bond et al., 2013). Globally, aerosols have helped to reduce the warming effect from greenhouse gases because of their net cooling effect on climate (Myhre et al., 2013). However, this influence is likely to be reduced over the coming decades as air pollution measures are implemented

around the world (Samset et al., 2018), as it is already the case in parts of Europe and North America (Collaud Coen et al., 2020). Therefore, in order to properly constrain global models, it is necessary to better characterize the atmospheric absorption by aerosols from observations. Among the atmospheric aerosols, black carbon (BC) stands out as phenomenologically different, being the most efficient light absorbing aerosol component and being responsible for the second most important contribution to positive climate forcing after carbon dioxide (Myhre et al., 2013). However, there are still large uncertainties related to the

radiative forcing of BC particles. In fact, the climate forcing potential of BC is influenced by BC properties which are strongly source and site dependent (Houghton, 2001; Ramanathan et al., 2001; Kirchstetter et al., 2004a; Ramanathan and Carmichael,

2008; Myhre et al., 2013; Bond et al., 2013; Liu et al., 2015). In addition to BC, atmospheric absorption by aerosol particles is also driven by specific organic compounds (e.g. from incomplete combustion, biomass smoldering, and secondary and biogenic sources) often referred to as Brown Carbon (BrC) and by mineral dust (e.g. Alfaro et al., 2004). Unlike BC, which

absorbs radiation in a wide range of wavelengths (from UV to infrared) with a wavelength independent refractive index, BrC and mineral dust refractive index increases at shorter wavelengths, close to the UV range (Kirchstetter et al., 2004b; Andreae and Gelencsér, 2006; Bergstrom et al., 2007; Laskin et al., 2015; Cappa et al., 2019). Therefore, having at disposal accurate absorption measurement techniques is crucial to determine particles light absorption which can afterwards be used in climate projections (Mengis and Matthews, 2020; Wang et al., 2020). Moreover, there is also the need of standard aerosol particles to

use as reference for quality assurance of absorption measurements, such as the recently developed flame-generated soot in Ess and Vasilatou (2019).

There are three main approaches in the literature to determine aerosol particles light absorption: by measuring the suspended particles in a cell, e.g. with photo-thermal interferometry or photo-acoustic techniques, and by either on-line or off-line filter-based photometer methods (e.g., Lin et al., 1973; Terhune and Anderson, 1977; Hansen et al., 1984; Stephens et al., 2003;

Moosmüller et al., 2009; Ajtai et al., 2010; Vecchi et al., 2014). Among the indirect methods for measuring absorption, the "subtraction method", which does not rely on a filter, calculates the absorption from the difference between extinction and scattering by suspended particles (Singh et al., 2014). However, this method can lead to large errors at large single scattering albedo (SSA) values when the extinction is dominated by scattering (Onasch et al., 2015). On-line measurement methodologies based on particle suspension, such as the photo-acoustic spectroscopy (PAS) (Ajtai et al., 2010), have the advantage of mea-

suring directly the absorption by particles suspended in a sampling cell avoiding filter-based artifacts. However, in the case of photo-acoustic spectroscopy measurements, the heating of the sample and the evaporation of coating materials on the sample may lead to higher detection limit and artifacts impairing the measurement accuracy (Lack et al., 2006; Linke et al., 2016). The photo-thermal interferometry (PTI) is an absorption measurement technique originally developed for measurements of trace gases that has also been applied to aerosol measurements (Lee and Moosmüller, 2020; Visser et al., 2020). However, the

aforementioned techniques have so far proved difficult to deploy in a field setting thus limiting their broader use in international measurements networks. Filter-based instruments (either on-line or off-line) rely on the sampling of aerosol particles collected in a filter matrix and on the measurement of the resulting change of light intensity with a photometer, either on the transmittance (Hansen et al., 1984; Bond et al., 1999; Drinovec et al., 2015), or on both transmittance and reflectance (Petzold and Schön-linner, 2004). This method is affected by artifacts resulting mainly from the effects that the filter has on the measurements.

Off-line in-house made filter based polar photometers, which measure both transmittance and reflectance, are deployed at some research centers. Examples are the MWAA (multi-wavelength absorption analyzer) deployed at University of Genoa (Massabò et al., 2013) and the PP_UniMI polar photometer deployed at University of Milan (Vecchi et al., 2014; Bernardoni et al., 2017). These methods can perform accurate absorption measurements by increasing the number of measuring angles (Massabò et al., 2013; Vecchi et al., 2014; Bernardoni et al., 2017) thus allowing an accurate determination of the filter artifacts.

The main advantage of the on-line filter-based methods is that these techniques are ease of use, allow for unattended operation, are relatively inexpensive and provide real-time data. For these reasons, these methods are widely used in international

networks such as the Global Atmosphere Watch (GAW, World Meteorological Organization) and the European Research Infrastructure for the Observation of Aerosol, Clouds and Trace Gases (ACTRIS; www.actris.eu). The most used filter-based instruments are the Aethalometer (Hansen et al., 1984; Drinovec et al., 2015), the Particle Soot Absorption Photometer (PSAP, Bond et al., 1999), the Continuous Light Absorption Photometer (CLAP; Ogren et al., 2017), and the Multi-Angle Absorption Photometer (MAAP, Model 5012, Thermo, Inc., USA; Petzold and Schönlinner, 2004). The measured mass concentration of light absorbing carbonaceous aerosol inferred via optical attenuation of light is referred to as equivalent BC (eBC; Petzold et al., 2013). The main artifacts affecting the light absorption measurements of these instruments are the multiple light scattering within the filter, the filter loading effect and the particle scattering correction (Liousse et al., 1993; Bond et al., 1999; Weingartner et al., 2003; Schmid et al., 2006; Collaud Coen et al., 2010; Lack et al., 2014). Algorithms for correcting these artifacts have been applied and their efficacy tested over the years (Weingartner et al., 2003; Arnott et al., 2005; Schmid et al., 2006; Virkkula et al., 2007; Collaud Coen et al., 2010; Virkkula et al., 2015). Currently, due to the described limitations of the filter-based photometers and other in-situ methods, no reference technique for measuring near-real time aerosol particles light absorption is available (Petzold et al., 2013; Lack et al., 2014).

The filter loading effect consists in the accumulation of particles and the consequent loss of sensitivity of the instrument with an increasing particle load (Bond et al., 1999; Weingartner et al., 2003; Lack et al., 2008; Moosmüller et al., 2009). The cross-sensitivity to scattering is the consequence of the multiple light scattering within the filter fibers and between particles and fibers, thus it is largely dependent on the single scattering albedo of the deposited aerosols. For the older Aethalometer model (AE31) the filter loading effect has been thoroughly studied and different methods for its quantification have been suggested. These methods use for example the discontinuity between the eBC concentration measurements before and after a filter spot is changed (Weingartner et al., 2003; Virkkula et al., 2007), or use the relationship between the eBC concentration and light attenuation (Park et al., 2010; Segura et al., 2014; Drinovec et al., 2015) to correct for filter loading effect. For the AE33 model the loading effect is corrected on-line using the dual-spot technology (Drinovec et al., 2015). In addition, the different physical and chemical properties of the collected particles influence particle optical properties, such as the backscatter fraction and the single scattering albedo (SSA), thus affecting also the multiple scattering of the collected particles and the filter loading effect (Weingartner et al., 2003; Lack et al., 2008; Virkkula et al., 2015; Drinovec et al., 2017). Among the on-line filter-based instruments, the Multi Angle Absorption Photometer (MAAP) also uses the measurements of light scattered by the blank and loaded filter in order to take into account both the loading effect and the aerosol particles multiple scattering. Consequently, the MAAP directly provides particle absorption coefficients similar to those obtained with other types of instruments (e.g. PAS; Petzold and Schönlinner, 2004; Petzold et al., 2005).

The multi-wavelength dual-spot Aethalometer software (AE33, Magee Scientific, Aerosol d.o.o. - Drinovec et al., 2015) corrects the loading effect on-line and directly implements the use of a correction factor (C) related to the multiple scattering within the filter matrix to convert the measured attenuation to an absorption coefficient. This C factor is generally assumed a priori, but it can be experimentally determined by using independent absorption measurements or by comparisons with other filter photometers(e.g. Weingartner et al., 2003; Arnott et al., 2005; Drinovec et al., 2015; Backman et al., 2017). For previous filter tapes and aethalometer versions different values of the multiple scattering parameter have been reported: for the AE31

quartz filter Weingartner et al. (2003) proposed a value of 2.14 which later on was recommended to be 3.5, i.e. larger by a factor of 1.64 (Müller, 2015; WMO, 2016); for the AE33 Drinovec et al. (2015) found a C of 1.57 for the Pallfex Teflon-coated glass fiber (TFE-coated glass, also known as M8020), which, after re-normalization using the factor 1.64, resulted in C=2.57.

Moreover, different experimental C factor values have been obtained ranging between 2.57-4.24 (Müller et al., 2011b; Drinovec et al., 2020; Laing et al., 2020; Valentini et al., 2020a; Bernardoni et al., 2020). In addition, the quartz filter for the AE31 and the TFE-coated glass filter for the AE33 have been found to feature a cross-sensitivity to scattering, $m_s$, ranging between 1 and 3% (Müller et al., 2011a; Müller, 2015; Drinovec et al., 2015; Zhang et al., 2018; Corbin et al., 2018; Laing et al., 2020; Drinovec et al., 2020). However, to the best of our knowledge, so far, no in-situ ambient measurements have been used for a

detailed characterization of the recommended new M8060 filter tape. Moreover, no sensitivity studies of the cross-sensitivity to scattering of the C factor have been reported and only very few studies dealt with the wavelength dependence of the C for either the previous and the current filter tapes.

The recent comparison between the MAAP and the off-line PP_UniMI polar photometer carried out by Valentini et al. (2020b) pointed to a possible measurement bias of the MAAP absorption coefficients. It is well established that the MAAP,

although limited to one measuring wavelength, is the most accurate filter-based on-line method available for the determination of the absorption coefficient (Petzold et al., 2005; Sheridan et al., 2005; Andreae and Gelencsér, 2006; Müller et al., 2011a). Therefore, it is often taken as the reference in inter-comparison exercises with other instruments, such as the AE33 e.g. in Backman et al. (2017). The discrepancy between MAAP and PP_UniMI reported by Valentini et al. (2020b) was mainly attributed to the value of the fraction of backscattered radiation set in the MAAP algorithm and directly measured by PP_UniMI

due to its high angular resolution which scans the whole scattering plane (resolution of 0.4 degrees in the scattering angle range 0-173°). Valentini et al. (2020b) also reported no differences between MAAP and PP_UniMI when the PP_UniMI was used with the same assumptions as those used in the MAAP (PaM as defined in Valentini et al., 2020b).

The main objective of this study is to characterize the C factor for different filter tapes used in AE33 instruments including the currently used M8060. To this aim, we compared the absorption coefficient measurements from the off-line PP_UniMI polar

photometer with the on-line MAAP and AE33 measurements performed at three measurement stations (urban background, BCN; regional background, MSY; and mountain-top, MSA) in the Western Mediterranean Basin (WMB). The novelty of this study relies in the the seasonal and diel variations analysis of the C factor, and especially, in the exploration of the cross-sensitivity to scattering of C and its relationship with the intensive optical properties of the collected particles at the three sites. This analysis allowed us to obtain both the multiple scattering parameter, $C_f$, and the cross-sensitivity to scattering, $m_s$,

constants for the M8060 filter currently used by the AE33 aethalometers. Moreover, we compared the results for the M8060 filter tape with the previously used TFE-coated glass filter tape (T60A20, also referred to as M8020) (Weingartner et al., 2003; Arnott et al., 2005; Drinovec et al., 2015). As aforementioned, the comparison between PP_UniMI and MAAP was reported in Valentini et al. (2020b) where data from BCN and MSY stations were also used to evaluate the performances of PP_UniMI vs MAAP. One of the main objectives of this study was using the multi-wavelength absorption coefficient measurements from

the off-line polar photometer extrapolated to the seven AE33 measurement wavelengths to study the wavelength dependence of the AE33 C factor at the three measurement sites.

## 2 Methodology

### 2.1 Measurement sites

Aerosols measurements were performed at Barcelona (BCN, urban background, $41°23'24.01''$N, $02°6'58.06''$E, 80 m a.s.l.),
Montseny (MSY, regional background, $41°46'46''$N, $02°21'29''$E, 720 m a.s.l.) and Montsec (MSA, mountain-top, $42°03'05''$N,
$00°43'46''$E, 1570 m a.s.l.) monitoring supersites (NE Spain). As shown later, these stations are characterized by aerosols with
different physical and chemical properties that influenced the differences obtained in the C values. A detailed characterization
of the three measurement stations can be found in previous works (e.g. Querol et al. (2001); Rodrıguez et al. (2001); Reche
et al. (2011) for BCN; Pérez et al. (2008); Pey et al. (2009); Pandolfi et al. (2011) for MSY; Pandolfi et al. (2014a); Ripoll
et al. (2014); Ealo et al. (2016) for MSA). Briefly, BCN station is located within the Barcelona metropolitan area of nearly
4.5 million inhabitants at a distance of about 5 km from the coast. MSY station is located in a hilly and densely forested area,
50 km to the N–NE of the Barcelona and 25 km from the Mediterranean coast. MSA station is located in a remote high-
altitude emplacement in the southern side of the Pre-Pyrenees at the Montsec d'Ares Mountain Range, at 140 km to the NW
of Barcelona and 140 km to the WNW of MSY. These supersites are part of the Catalonian Air Quality Monitoring Network,
and are part of ACTRIS and GAW networks. Aerosol optical properties at the three sites were measured following standard
protocols (WMO/GAW, 2016).

The area of study is characterized by high concentrations of both primary and secondary aerosols, especially in summer
(Rodríguez et al., 2002; Dayan et al., 2017; Rivas et al., 2020; Brean et al., 2020) from diverse emission sources. Anthropogenic
emissions from road traffic, industry, agriculture, and maritime shipping, among others, strongly contribute to the air quality
impairment in this region (Querol et al., 2009b; Amato et al., 2009; Pandolfi et al., 2014b). Moreover, the Mediterranean Basin
is also highly influenced by natural sources, such as mineral dust from African deserts and smoke from forest fires (Bergametti
et al., 1989; Querol et al., 1998; Rodrıguez et al., 2001; Lyamani et al., 2006; Mona et al., 2006; Koçak et al., 2007; Kalivitis
et al., 2007; Querol et al., 2009b; Schauer et al., 2016; Ealo et al., 2016; Querol et al., 2019, among others).

### 2.2 Aerosol characterization

#### 2.2.1 Aerosol absorption and eBC measurements

The on-line aerosol absorption coefficient, $b_{abs}$, was measured at the three sites with a multi angle absorption photometer
(MAAP, Model 5012, Thermo Inc., USA, Petzold and Schönlinner, 2004). This instrument derives the absorption coefficient
at 637 nm (Müller et al., 2011a) and eBC concentration using a radiative transfer model from the measurements of transmis-
sion of light through the filter tape and backscattering of light at two different angles. Black carbon, eBC, and attenuation
measurements, $b_{atn}$, were also performed with the AE33 multi-wavelengths aethalometer (model AE33, Magee Scientific,
Aerosol d.o.o. Drinovec et al., 2015). The AE33 is based on the measurement at 7 different wavelengths (370, 470, 520, 590,
660, 880, and 950 nm) of the transmission of light through two sample spots with different flows and particle loading relative

to the reference spot. It derives the eBC concentration and the attenuation coefficients by applying eqs. (1) and (2), respectively, following Drinovec et al. (2015):

$$eBC = \frac{S \cdot (\Delta \text{ATN}_1/100)}{F_1(1-\zeta) \cdot \sigma_{abs} \cdot C(1-k\Delta \text{ATN}_1) \cdot \Delta t};\qquad(1)$$

$$b_{atn} = \frac{S \cdot (\Delta \text{ATN}_1/100)}{F_1(1-\zeta) \cdot (1-k\Delta \text{ATN}_1) \cdot \Delta t},\qquad(2)$$

where S is the filter surface area loaded with the sample, $F_1$ the volumetric flow of the spot 1, $\zeta$ the lateral airflow leakage, $\sigma_{abs}$ the mass-absorption cross-section, k the loading factor parameter and $\Delta \text{ATN}_1$ the variation of attenuation of light of the filter tape loaded with the sample of the spot 1, $\text{ATN}_1$, during the measurement timestamp $\Delta t$.

The Aethalometer absorption coefficient can be derived by dividing the attenuation coefficient (eq. 2) by the multiple scattering parameter C of the filter tape

$$b_{abs} = \frac{b_{atn}}{C}\qquad(3)$$

Off-line multi-wavelength particle absorption coefficients were obtained using the PP_UniMI polar photometer (Vecchi et al., 2014; Bernardoni et al., 2017) measurements on the MAAP filter spots. 85 filter spots collected at BCN in the period October 2018 - June 2019, 126 filter spots collected at MSY between June - December 2018 (Valentini et al., 2020b), and 122 filter spots collected at MSA between June and November 2018 were analyzed. The time elapsed between the MAAP measurements and the MAAP spots analysis with the PP_UniMI in Milan varied between one year and one month. Once selected and cut, each MAAP spot was stored in a petri dish in a fridge and then sent to Milan. We assumed that there were no major particle losses affecting the measured optical properties, although some volatile compounds could have been evaporated over the period. The PP_UniMI measures the transmitted and scattered radiation at 4 wavelengths (405, 532, 635 and 780 nm) in a range of scattering angles from $0°$ to $173°$ with a resolution down to $0.4°$ and applies a radiative transfer model to derive the absorption coefficients. The PP_UniMI working principle and the detailed analysis of the inter-comparison between the MAAP and PP_UniMI for different measurement sites, including BCN and MSY, was reported in Vecchi et al. (2014)Bernardoni et al. (2017) and in Valentini et al. (2020b). As mentioned before, in these studies no differences were observed between MAAP and PP_UniMI when the latter was used as a MAAP (PaM), i.e. using a data inversion with similar assumptions as those performed in the MAAP.

Here we obtain the wavelength dependent attenuation coefficients $b_{atn}(\lambda)$ derived exclusively from the AE33 measurements by multiplying the eBC concentrations provided by the AE33 (eq. 4) by the default wavelength independent instrumental filter constant $C_0$ from the AE33 setup file (1.57 for the TFE-coated glass fiber tape T60A20, also referred to as M8020; and 1.39 for the M8060 filter tape),

$$b_{atn}(\lambda) = eBC(\lambda) \cdot \sigma_{abs}(\lambda) \cdot C_0 = \frac{S}{F} \frac{\Delta \text{ATN}(\lambda)}{\Delta t} \cdot f(ATN, \lambda)\qquad(4)$$

where $f(ATN, \lambda)$ is the function which contains all the corrections, i.e. filter loading and leakage, which are performed by the AE33 for each wavelength (Drinovec et al., 2015). Note that the new filter tape M8060 structurally differs from the old

filter tape M8020 in filter fibers material, thickness and density, thus leading to different $C_0$ values (details can be found in the following online document from Magee Scientific: https://mageesci.com/tape/Magee_Scientific_Filter_Aethalometer_AE_Tape_Replacement_discussion.pdf).

Then, we determined the average and seasonal multiple scattering factor C both as the ratio between the AE33 attenuation coefficients and the absorption coefficients $b_{abs}(\lambda)$ measured by the MAAP and the PP_UniMI (eq. 5), and also by applying a Deming regression between the AE33 attenuation coefficients and the MAAP absorption coefficients for the overall average values for each filter tape.

$$C(\lambda) = \frac{b_{atn}(\lambda)}{b_{abs}(\lambda)} \tag{5}$$

This value of the multiple scattering parameter $C(\lambda)$ is the value derived from the experimental comparison of different instruments, contrasting the default instrumental constant value $C_0$. The data availability at BCN station ranged between 2016 and 2020, at MSY and MSA data was measured from 2013 to 2020. Different AE33 filter tapes were used during these periods at the three stations as shown in Fig. S1.

### 2.2.2 Aerosol scattering measurements

On-line particle total scattering ($b_{sp}$) and hemispheric backscatter ($b_{bsp}$) coefficients were measured on-line at the three sites with LED-based integrating nephelometers (Aurora 3000, ECOTECH Pty Ltd, Knoxfield, Australia) operating at three wavelengths (450, 525 and 635 nm). Calibration of the nephelometers was performed three times per year using $CO_2$ as span gas while zero adjusts were performed once per day using internally filtered particle-free air. The RH threshold was set by using a processor-controlled automatic heater inside the Aurora 3000 nephelometer to ensure a sampling RH of less than $40\%$ (GAW, 2016). $\sigma_{sp}$ coefficients were corrected for non-ideal illumination of the light source and for truncation of the sensing volumes following the procedure described in Müller et al. (2011b).

### 2.3 Data treatment and conceptual model

The different analyses performed herein were performed considering the absorption coefficients provided either by the MAAP or the PP_UniMI as reference absorption measurements depending on either time resolution and coverage, or on the measurement availability at several wavelengths. The AE33 and MAAP data (provided with high temporal resolution) were used to study the seasonal variations and the cross-sensitivity to scattering of the C factor. The AE33 and PP_UniMI data (provided with low temporal resolution but at different wavelengths) were used to determine the wavelength dependence of the C factor.

### 2.3.1 Average, seasonal values analysis and cross-sensitivity to scattering analysis

As aforementioned, the seasonal analysis of the C factor, its average values and the study of its cross-sensitivity to scattering were performed using the long high-time resolution dataset from the MAAP and AE33 measurements at the three measurement sites. For this, we applied eq. (5) using the absorption coefficient from the MAAP and the AE33 attenuation coefficient

extrapolated to the 637 nm wavelength of the MAAP through the Ångström exponent obtained from the AE33 measurements at 7 wavelengths.

The cross-sensitivity to scattering which, as shown later, can strongly affect the C factor values, is neglected in AE33 applications where it is generally assumed that the measured light attenuation is only due to the absorption of light by the collected particles (eqs. 1-2). Moreover, it is also generally assumed that the multiple scattering by particles is sample independent, or constant, and can be taken into account by introducing the multiple scattering correction factor C (Drinovec et al., 2015). However, this assumption is a first approximation, since the attenuation of transmitted light is also due to the scattering of light by the collected particles (Bond et al., 1999; Arnott et al., 2005). Taking this dependence into account and following Arnott et al. (2005) and Segura et al. (2014), we parameterized the light attenuation coefficient as

$$b_{atn} = \frac{S}{F} \frac{\Delta ATN}{\Delta t} \cdot f(ATN) + m_s \cdot b_{sp}, \tag{6}$$

to obtain the relationship between the absorption, attenuation and scattering coefficients

$$b_{abs} = \frac{b_{atn}}{C_f} - m \cdot b_{sp}, \tag{7}$$

The cross-sensitivity to scattering, which is denoted by the constant $m_s$, is related with $m$ through $m = m_s/C_f$. Here $C_f$ refers to the filter multiple scattering parameter, that is a value (possibly wavelength dependent) that depends only on filter properties. If we rearrange eq. (7) by expressing the scattering coefficient through the single scattering albedo, we obtain the dependence of the absorption as a function of SSA (eq.8), similarly to eq. 17 in Schmid et al. (2006). The measured multiple parameter, C, affected by the cross-sensitivity to scattering can be expressed as shown in eq. (9).

$$b_{abs} = \frac{b_{atn}}{C_f} \cdot \frac{1}{1 + m \cdot \frac{SSA}{1 - SSA}} \tag{8}$$

$$C = C_f \left(1 + m \cdot \frac{SSA}{1 - SSA}\right) = C_f + m_s \cdot \frac{SSA}{1 - SSA} \tag{9}$$

The effective multiple scattering parameter, C, depends on the physical properties of collected particles. By comparing data from different instruments (AE33, MAAP, and nephelometer) we were able to parameterize the cross-sensitivity of the C to scattering (eq. 8). Eq.9 shows that the actual AE33 cross-sensitivity to scattering is more pronounced when the measured aerosol particles have higher SSA, whereas for particles with lower SSA eq.9 converges to eq. 5.

By analyzing the dependency of the effective multiple scattering parameter C with the SSA we obtained the experimental fit constants ($C_f$ and $m_s$) that describe the relationship between C and SSA. Furthermore, we will present in Section 3.1 how the cross-sensitivity to scattering of C depended on some intensive aerosol particle optical properties that strongly depend on aerosol particles size distribution and chemical composition (Figs. S3-S5).

The AE33 data treatment applied to obtain the C seasonality and the cross-sensitivity to scattering included a pre-process filtering method following the approach suggested in Springston and Sedlacek (2007) and Backman et al. (2017). This filtering method consists on setting a threshold value for the measured attenuation variation, $\Delta ATN_1$, high enough so that the signal-to-noise ratio is large; herein we have used a fixed value of 0.01. As can be deduced from eq. (1), the faster the fixed $\Delta ATN_1$

is reached, the shorter is the period $\Delta t$, implying therefore a higher eBC concentration value during the same period. The method we employed determines the period at which the $\Delta \mathrm{ATN}_1$ step was reached, and recalculated the eBC concentration for this $\Delta t$. As a consequence of this eBC re-calculation, we filtered out the noise resulting from very small values close to the detection limit of the instrument while maintaining the higher eBC values measured without introducing a bias to the measurements as is the case when averaging. With the aim to study the seasonality of the C factor and its cross-sensitivity

to scattering, we averaged $b_{abs,MAAP}$ and $b_{sp}$ coefficients to match the corresponding AE33 variable timestamp, $\Delta t$, which ranged approximately between 3 and 14 min (cf. Fig. S2). Moreover, the time granularity of the measurements varied between 1 to 5 minutes, depending on the software used for data logging (see Table S1). Given the length of the measurement periods, we assumed that the AE33 filter tapes considered here were characterized under a wide range of aerosol particle properties typically observed at the measurement stations and that the non-simultaneity of AE33 measurements with the two filter tapes

did not prevent the comparison between the obtained C.

### 2.3.2   Wavelength dependence analysis

To study the wavelength dependence of the C factor we compared the absorption coefficients at several wavelengths measured with the PP_UniMI with the attenuation coefficients obtained from the AE33 (eq. 5). Since the off-line PP_UniMI measurements were performed on the MAAP spots, the measured attenuation and scattering coefficients from AE33 and nephelometer,

respectively, were averaged over the timestamp of each one of the selected MAAP spots. The absorption coefficients from the PP_UniMI were inter/extrapolated to the seven AE33 wavelengths using the attenuation Ångström exponent, obtained through a log-log fit from the PP_UniMI absorption measurements.

Valentini et al. (2020b) reported that the MAAP overestimates the absorption coefficient compared to the PP_UniMI. For BCN and MSY Valentini et al. (2020b) reported a MAAP overestimation of $18\%$ and $21\%$, respectively. By applying the same

methodology as in Valentini et al. (2020b) we obtained a difference between MAAP and PP_UniMI for MSA of $19\%$ (Fig. A1) similar to the biases obtained for BCN and MSY. For this reason, Valentini et al. (2020b) also studied the comparison between MAAP and PP_UniMI using for the PP_UniMI data inversion the same assumptions as those performed in the MAAP (PaM approach) and reported a 1:1 correlation between the two instruments. Given that most of the aethalometer C values reported in literature were obtained by comparing AE33 attenuation measurements and MAAP absorption measurements, we report here

also the median C values obtained comparing the AE33 with the PP_UniMI (Table S2) and with PaM (Table S3).

## 3   Results

### 3.1   Multiple scattering parameter cross-sensitivity to scattering

The cross-sensitivity to scattering of the C factor at the three stations was obtained by analyzing the relationship between the multiple scattering parameter C (at 637 nm) and the measured SSA (eq. 9).

**Table 1.** AE33 multiple scattering parameter C for some measurement stations (included BCN, MSY and MSA) and cross-sensitivity to scattering for BCN, MSY and MSA station compared to literature values for AE33 TFE-coated glass (M8020). Different approaches, as aforementioned in Section 3.2, have been used to obtain the factor C. Since the literature values are obtained through either one of the methods, we include these vales in its corresponding column (C or $C_{Deming}$).

| Site | Characteristics | Filter type | Reference | C | $C_{Deming}$ | $C_f$ | $m_s$ (%) |
|---|---|---|---|---|---|---|---|
| **Barcelona** | Urban background | M8020 | This study | **2.29 ± 0.49** | **1.99 ± 0.02** | - | - |
| | | M8060 | This study | **2.44 ± 0.57** | **2.20 ± 0.02** | **2.50 ± 0.02** | **1.6 ± 0.3** |
| Leipzig | Urban background | M8020 | Müller (2015) | 3.2 | | | |
| | | M8020 | Bernardoni et al. (2020) | | 2.78 | | |
| Rome | Urban background | M8060 | Valentini et al. (2020a) | 2.66 | | | |
| Klagenfurt | Urban background | M8020 | Drinovec et al. (2020) | 1.57 | | | |
| **Montseny** | Regional background | M8020 | This study | **2.29 ± 0.46** | **2.05 ± 0.02** | **2.21 ± 0.01** | **1.8 ± 0.1** |
| | | M8060 | This study | **2.23 ± 0.30** | **2.13 ± 0.01** | **1.96 ± 0.01** | **3.0 ± 0.1** |
| **Montsec d'Ares** | Mountain-top | M8020 | This study | **2.36 ± 0.59** | **2.21 ± 0.03** | **1.96 ± 0.02** | **3.4 ± 0.1** |
| | | M8060 | This study | **2.51 ± 0.71** | **2.05 ± 0.02** | **1.82 ± 0.02** | **4.9 ± 0.1** |
| Mt. Bachelor | Mountain-top | M8020 | Laing et al. (2020) | 4.24 | | | |

The SSA was obtained independently at 637 nm using simultaneous MAAP and multiple-wavelength integrating nephelometer data. C was obtained through eq. (5) from the AE33 attenuation coefficient, extrapolated at 637 nm using the AAE from AE33, and the MAAP absorption coefficients at 637 nm. The analysis was performed by binning the SSA data using Freedman and Diaconis (1981) criteria, and then averaging the obtained C values within each SSA bin. Binned data were then fitted following (9) to obtain the experimental values of both $C_f$ and $m_s$.

Figure 1 and Table 1 show the results of the fit for BCN, MSY and MSA for both M8020 and M8060 filter tapes. Moreover, Table 1 compares the C values obtained here with those reported in literature for the M8020 filter tape. For M8020, we calculated a constant $C_f$ of 2.21 ± 0.01 and a cross-sensitivity to scattering, $m_s$, of 1.8 ± 0.1 at MSY, and of 1.96 ± 0.02 and 3.4 ± 0.1 % at MSA. For the M8060 filter tape, the fit yielded a multiple scattering constant $C_f$ of 2.50 ± 0.02 and a cross-sensitivity to scattering of 1.6 ± 0.3 % at BCN, a $C_f$ of 1.96 ± 0.01 and a $m_s$ of 3.0 ± 0.1 % at MSY, and a constant $C_f$ of 1.82 ± 0.02 and a $m_s$ of 4.9 ± 0.1 % at MSA.

As a consequence of the cross-sensitivity to scattering, we can appreciate in Fig. 1 a clear increase of C with increasing SSA with an up to 3-fold increase of C for SSA>0.90-0.95 depending on the station and filter tape considered. The cross-sensitivity to scattering was evident for both filter tapes at the regional (MSY) and mountain (MSA) stations where the probability of measuring SSA higher than 0.90-0.95 was high (57-70% of the data in Fig. 1). Conversely, at the urban site (BCN), where the SSA was on average lower (12% of SSA data was above 0.90), a lower cross-sensitivity to scattering was observed. This significant increase of the C factor at high SSA, if not accounted for, can lead to a large overestimation of both eBC concentrations and absorption coefficients from Aethalometer instruments. This effect can have a larger impact at sites where

high SSA values are typically observed as remote arctic sites and mountain-top sites (Collaud Coen et al., 2004; Gyawali et al., 2009; Andrews et al., 2011; Pandolfi et al., 2014a, 2018; Schmeisser et al., 2018; Ferrero et al., 2019; Laj et al., 2020), as well as in places where increasing or decreasing trends of SSA have been observed (Collaud Coen et al., 2020). This cross-sensitivity to scattering of the filter explains the higher C factors obtained on average at these types of sites (Table 1) and suggests the need of using either a site-specific C, or a C that takes into account the SSA measured by an independent absorption method. Given its impact on the absorption coefficient, this effect needs to be taken into account for climate studies.

In order to further characterize the observed cross-sensitivity to scattering, we explored how the variations of C with SSA depended on different aerosol particle intensive optical properties, namely AAE (Fig. S3), backscatter fraction (BF; Fig. S4) and single scattering albedo Ångström exponent (SSAAE; Fig. S5). We found that large C values, and high SSA, were often obtained when the sampled aerosol composition was dominated by mineral dust during Saharan dust outbreaks, as demonstrated by the occurrence of negative SSAAE at high SSA (Fig. S5). In fact, Saharan dust outbreaks, which are common in the WMB (Escudero et al., 2005; Querol et al., 2004, 2009b, a; Ealo et al., 2016; Querol et al., 2019; Yus-Díez et al., 2020), have the potential to increase the SSA above the average velues especially at the regional (MSY) and remote (MSA) stations (e.g. Pandolfi et al., 2014a). In prior studies, negative values of the SSAAE have been associated with an aerosol mixture dominated by mineral dust (Collaud Coen et al., 2004; Ealo et al., 2016; Yus-Díez et al., 2020). Moreover, we observed that high C values (for SSA>0.95) were also associated with AAE values higher than around 1.5 (cf. Fig. S3) thus indicating a relatively higher absorption efficiency of the collected particles in the UV, consistent with the presence of either dust or brown carbon (BrC) particles (Kirchstetter et al., 2004b; Chen and Bond, 2010; Zotter et al., 2017; Forello et al., 2019, 2020). Furthermore, low BF values, indicative of the predominance of large particles, were also on average associated with high SSA values (cf. Fig. S4). Note that the dependence of the C vs. SSA on the aforementioned intensive optical properties was not clearly observed in BCN, where, at least for the period under study, local pollution masked the effects of coarse dust particles on the measured intensive optical properties and on SSA which kept values lower than around 0.90-0.95. The observed dependency of C on aerosol particle intensive optical properties demonstrated that both particle size distribution and chemical composition can affect the reported C vs. SSA relationships.

### 3.2 Multiple scattering correction factor: Average values and seasonal variation

Here we present the average values and the seasonal cycle of the C factor calculated at 637 nm at BCN, MSY and MSA. We analyzed the multiple scattering parameter C values both through a Deming regression, taking into account the measurement error of the MAAP (12%; Petzold and Schönlinner, 2004) and of the AE33 (15%; Zanatta et al., 2016; Rigler et al., 2020), and by calculating the median value of the C factor density distribution. The uncertainties of the C factor were derived as either the methodological error from the regression slope of the Deming fit, or as the half-width at half maximum (HWHM) of the density distribution of the C factor. We present here the results from both the aforementioned methods because both methods have been reported in literature (e.g. Backman et al., 2017; Bernardoni et al., 2020, ; c.f. Table 1 in this work).

The density distribution of the C factor obtained from the ratio (with a variable time resolution, as aforementioned in Sect. 2.3.1), showed a quasi-Gaussian distribution at the three measurement sites with a small tail toward higher C values (Fig. 2).

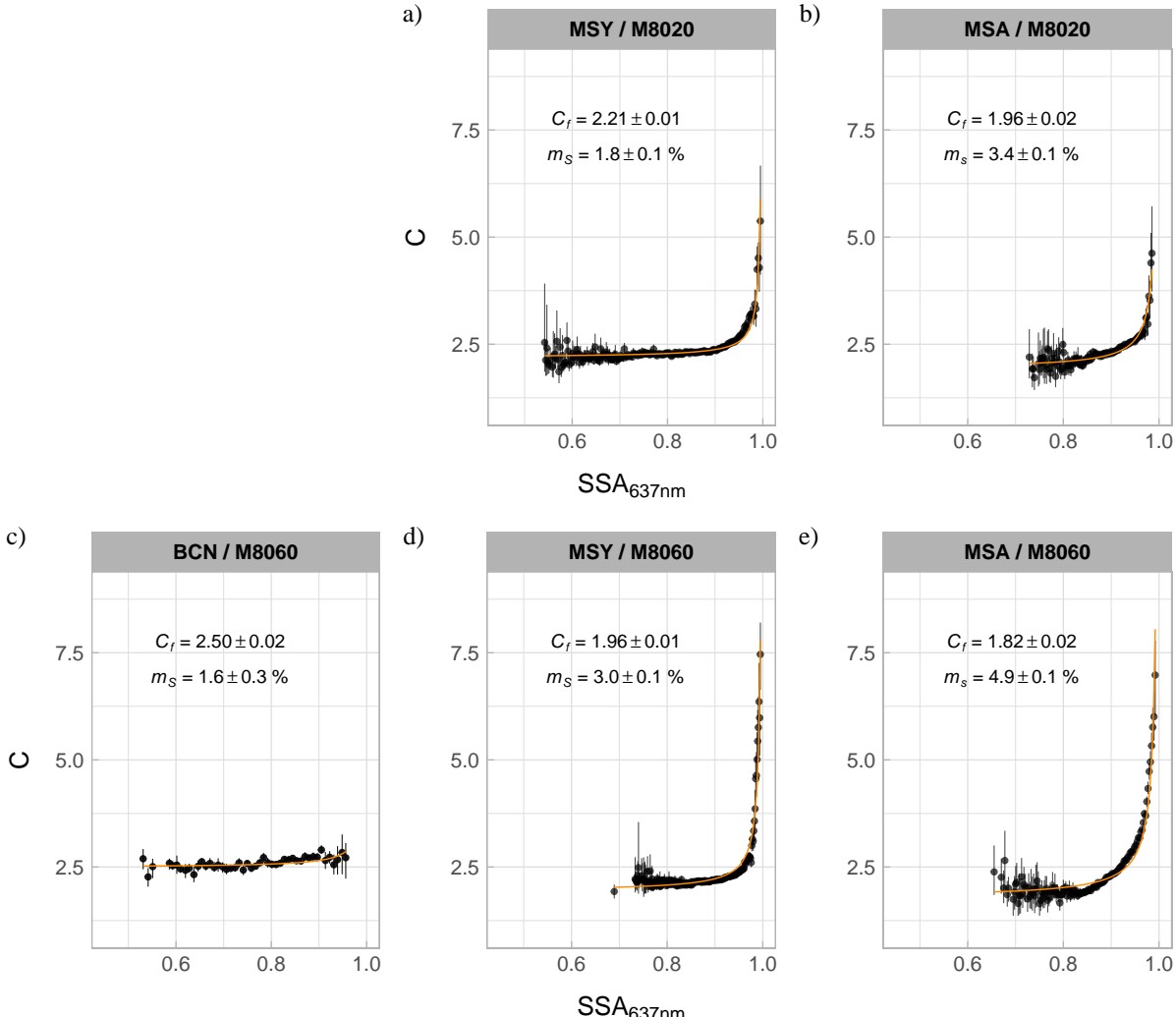

**Figure 1.** Cross-sensitivity to scattering analysis for TFE-coated glass tape (also known as M8020, upper panel) and M8060 filter tape (lower panel) for BCN (c), MSY (a,d) and MSA (b,e) stations obtained by attenuation coefficients from the AE33, absorption coefficients from the MAAP photometer and scattering coefficients from the integrating nephelometer. Each data point represents the mean, and the vertical bars the first and third quartile for each bin. Multiple scattering constant, $C_f$ and cross-sensitivity to scattering, $m_s$, are determined by fitting eq. (9) to the binned data.

The median values of the C factor for the M8020 filter tape were $2.29 \pm 0.48$, $2.29 \pm 0.46$, $2.36 \pm 0.59$ for BCN, MSY and MSA, respectively. These values were on average similar or slightly lower (with differences less than a 7%) compared to the median C values obtained for the M8060 filter tape of $2.44 \pm 0.57$, $2.23 \pm 0.30$, and $2.51 \pm 0.71$ (cf. Table 1). The Deming regression fit results (Fig. S6) showed C values of $1.99 \pm 0.02$, $2.05 \pm 0.02$, and $2.21 \pm 0.03$ (at BCN, MSY and

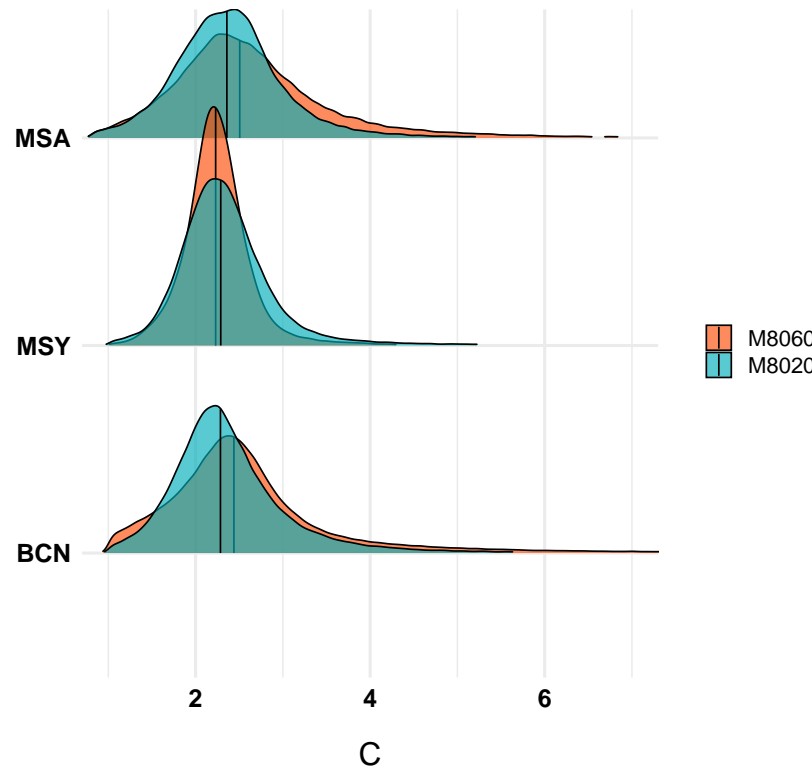

**Figure 2.** Density distribution of the C factor for each filter type, M8020 and M8060, as obtained through eq. (5) using both attenuation coefficient from the AE33 and the absorption coefficient from the MAAP. The vertical line represents the median value of each distribution.

MSA, respectively) for the M8020 which were slightly lower (with differences <10%) compared to the C values of $2.20 \pm 0.02$, $2.13 \pm 0.01$, and $2.05 \pm 0.02$ obtained for the M8060. Note that the uncertainties from the Deming regression were lower compared to the uncertainties derived as HWHM of the distributions because the Deming regressions were performed using binned data (cf. Fig. S6). This also was the likely explanation for the lower C values on average obtained with the Deming regression compared to the median values of the density distribution. The difference of the C values between both methods ranged between 4-18% depending on the filter tape/measurement station considered (cf. Table 1). However, both methods were consistent and provided higher C factor for the M8060 than for the M8020 filter tape.

As reported in Table 1, overall, higher C values were found at MSA, where both the SSA and the cross-sensitivity of the filter tape to scattering were higher compared to MSY and BCN (cf. Figs. 1 and S7). The C values for the AE33 M8020 and M8060 filter tapes obtained at urban background stations in Rome (Valentini et al., 2020a) and Leipzig (Müller, 2015; Bernardoni et al., 2020) were in the same range as those found in this work for BCN (Table 1).

Figure 3 shows the seasonal variability of the C factor for the TFE-coated glass and M8060 filter tapes at the three stations. It can be appreciated the large variability of the obtained C parameters (cf. Fig. 3) at the three sites during all the seasons, consistent with the width of the C factor density distribution(Fig. 2) and the SSA seasonal evolution variability (Fig. S7).

On average, an increase of C was observed at MSY and MSA in summer (JJA) for both filter tapes. This increase was likely driven by a greater influence of diurnal processes and the impact of the atmospheric boundary layer (ABL) during the warm months at these two elevated stations, and by changes in the chemical and physical properties of collected particles in summer compared to winter (DJF). In fact, spring and summer seasons in the WMB are characterized by a high frequency of Saharan dust outbreaks (e.g. Pey et al., 2013; Yus-Díez et al., 2020) and formation of high concentrations of secondary organic aerosols and secondary sulfate particles (e.g. Ripoll et al., 2015) which in turn increase the particle scattering efficiency and the SSA in summer compared to winter (Pandolfi et al., 2011). Although dust particles can absorb radiation (e.g. Sokolik and Toon, 1999; Di Biagio et al., 2019), the effect of Saharan dust outbreaks at the measurement stations considered here was to increase the SSA (at 637 nm) over the average values. In fact, as shown by Pandolfi et al. (2014a), both scattering and absorption increased at MSY and MSA during Saharan dust outbreaks, but the resulting SSA was higher compared to other atmospheric scenarios typical of the area under study. Therefore, the higher C values observed during Saharan dust outbreaks were coherent with an increase of SSA over the threshold above which the C sharply increased (cf. Fig. 1, S3, S4 and S5). An increase of the C when dust particles are deposited on the filter tape was also reported by Di Biagio et al. (2017) for the AE31 aethalometer. Di Biagio et al. (2017) reported C values for dust particles by generating particles by mechanical shaking of dust samples from different desert soils using AE31 and MAAP measurements, and reported C values between 3.6 and 3.96 for Saharan desert soils (Table 2 of Di Biagio et al. (2017)).

As shown in Sect. 3.1, high SSA increased the C values, and, consequently, the C seasonality was affected, to some degree, by the SSA seasonality. In fact, Fig. S7 in the supplementary material shows that the seasonal evolution of the SSA at MSY and MSA mirrored quite well the seasonal evolution of the C, with an increase of both C and SSA toward the warm season. In BCN, the inter-season variability of both C and SSA was less pronounced and the C remained fairly constant during the different seasons. Exception was in the winter period (DJF) when both C (M8060) and SSA showed minima. Nevertheless, the variability within each season was the largest in BCN, due to a higher variability of the SSA values at this station within each season compared to MSY and MSA (Fig. S7). The relationship between C and SSA can be also observed in Fig. S8, where the diel cycles of both C and SSA were reported. In BCN, both C and SSA showed two relative minima in the morning and in the afternoon, mirroring the traffic rush hours. At MSY, the sea-breeze-driven transport of pollutants in the afternoon caused a reduction of both SSA and C. Conversely, at MSA both C and SSA showed less variability in the diel cycles and less similarity was observed. Note that the similarities commented above between the diel/seasonal cycles of C and SSA were more or less evident depending on the season/station considered. In fact, we have shown in Fig. 1 that high SSA (> 0.90-0.95) can strongly affect the C values, but less dependency between C and SSA was observed for lower SSA thus also contributing to mask the similarities between C and SSA reported in Figs. 3, S7, and S8 which were obtained averaging all available data, including C values at lower SSA.

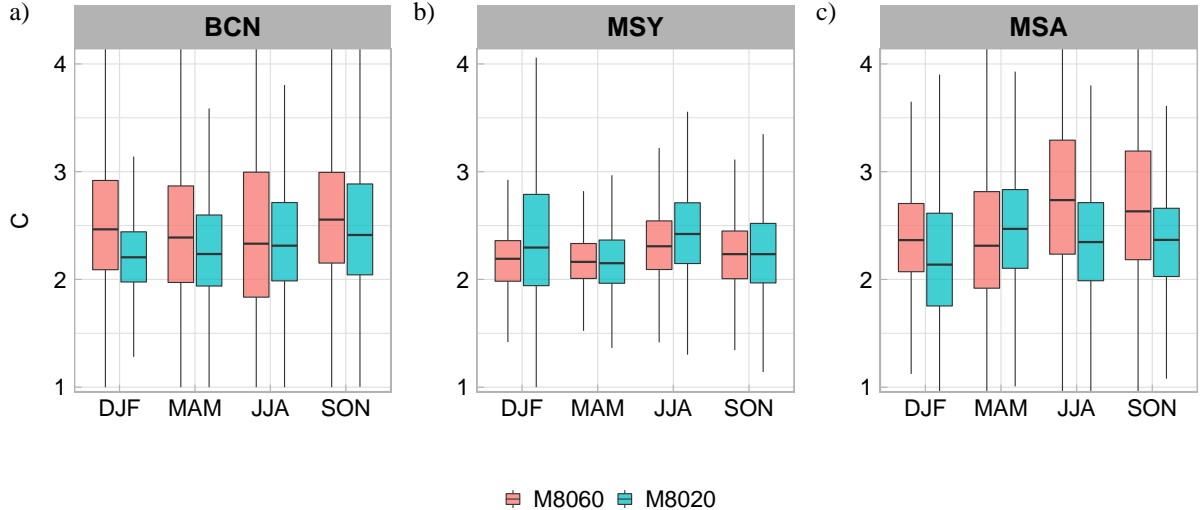

**Figure 3.** Seasonal evolution of the C factor at a) BCN, b) MSY and c) MSA measurement stations for both TFE-coated glass (M8020) and M8060 filter tapes. The box plot boxes show the range between the first and third quartile (IQR) with the median value for each season distribution represented by the inner line; the maximum whisker length is proportional to 1.5 times the third and first quartile difference, or inter-quartile range (1.5 · IQR).

### 3.3 Wavelength dependence analysis from the PP_UniMI vs AE33 comparison

The spectral dependence of the AE33 C factor, C($\lambda$), was studied at the three stations by comparing the attenuation coefficients, $b_{atn}$, from AE33 at seven different wavelengths with the absorption coefficients, $b_{abs}$, from the PP_UniMI. To this aim, the PP_UniMI absorption coefficients were inter/extrapolated to the seven AE33 wavelengths using the Absorption Ångström Exponent (AAE) obtained from the original PP_UniMI measurements. The obtained mean AAE were 1.12 ± 0.17, 1.29 ± 0.24, and 1.35 ± 0.18 for BCN, MSY, and MSA stations, respectively, with an increase from the urban (BCN) to the regional (MSY) and remote (MSA) sites due to the increase in the relative importance of non-fossil BC sources (e.g. biomass burning) and Saharan dust at the remote sites compared to BCN.

Fig. 4 shows that at the urban (BCN) and regional (MSY) stations the C factor did not present a statistically significant dependence with the wavelength. However, Fig. 4c shows that at the remote MSA station the multiple scattering parameter C presented a statistically significant increase between 370 nm (C=3.47) and 950 nm (C=4.03) (cf. Table S2). The observed increase of the C factor with wavelength affects the absorption coefficients derived from the AE33 attenuation measurements and, consequently, can affect all the intensive optical parameters such as AAE, SSA and SSAAE which can be derived from the multi-wavelength AE33 absorption measurements and scattering coefficient measurements. Moreover, a wavelength-dependent C factor can impair aethalometer based BC source apportionment analysis, such as the Aethalometer model, used to determine the contribution from fossil fuels vs biomass burning emissions (Sandradewi et al., 2008). Contradictory results have been reported in literature about the spectral dependence of C for older versions of aethalometer (model AE31). For example,

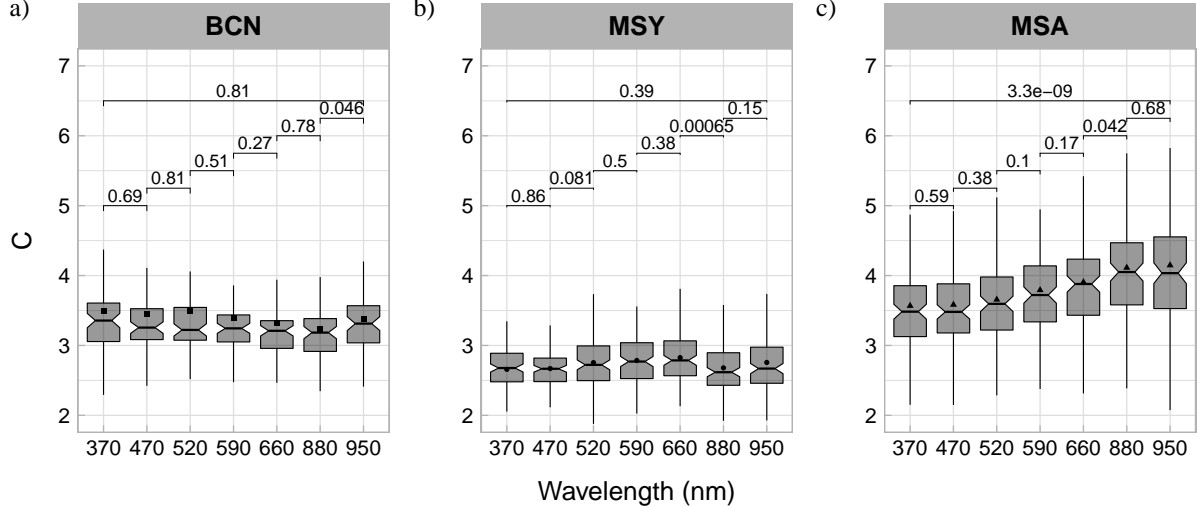

**Figure 4.** Wavelength dependence of C at BCN, MSY and MSA comparing $b_{atn}$ from the AE33 measured at each wavelength and $b_{abs}$ inter/extrapolated to the same wavelength from the PP_UniMI. Box plots have been obtained as in Fig. 3 with the addition of the mean value of the distribution for each wavelength represented by a marker. The values above the box plots between adjacent wavelengths and between 370 and 950 nm wavelength box plots show the obtained p-values, with p<0.05 meaning a statistically significance difference.

Weingartner et al. (2003) found strong indication of the independence of C with wavelength, and neither Segura et al. (2014) found any wavelength dependence of the multiple scattering parameter C with the wavelength. Conversely, Bernardoni et al.
(2020) observed a decrease of the C factor with wavelengths, although it was not statistically significant.

As can be appreciated by comparing Figs. 2, 3 and Fig. 4, the multiple scattering correction factors obtained using the PP_UniMI reference instrument were larger than those obtained with the MAAP as a consequence of the offset in the absorption measurements between MAAP and PP_UniMI. A detailed discussion of this offset can be found in Fig. A1 and in Fig. 2 in Valentini et al. (2020b).

Hereafter, we propose a possible explanation for the different spectral dependencies found for the C at the measurement sites considered here. We have shown in Section 3.1 that, independently from the measurement station considered, the cross-sensitivity to scattering can strongly increase the C for SSA values above an upper threshold. To explore if the SSA can also affect the C wavelength dependence, we studied the wavelength dependence of the C for SSA values above and below the site-dependent SSA thresholds. Figure 5 shows the comparison between the C factor at MSY and MSA for SSA above (high
SSA), and below (low SSA) the SSA thresholds of 0.95 and 0.9, respectively, for MSY and MSA (cf. Fig. 1). Fig. 5 shows that at MSA there was a statistically significant increase of C with the wavelength for SSA>0.90, whereas no statistically significant increase was observed for SSA<0.90. For this specific analysis, based on the PP_UniMI off-line measurements, 86% of SSA values at MSA (68 samples out of 79) were above the SSA threshold of 0.95. At MSY, only 1 sample out of 126 was characterized by SSA value higher than the SSA threshold of 0.95, thus preventing a robust statistical analysis of the C

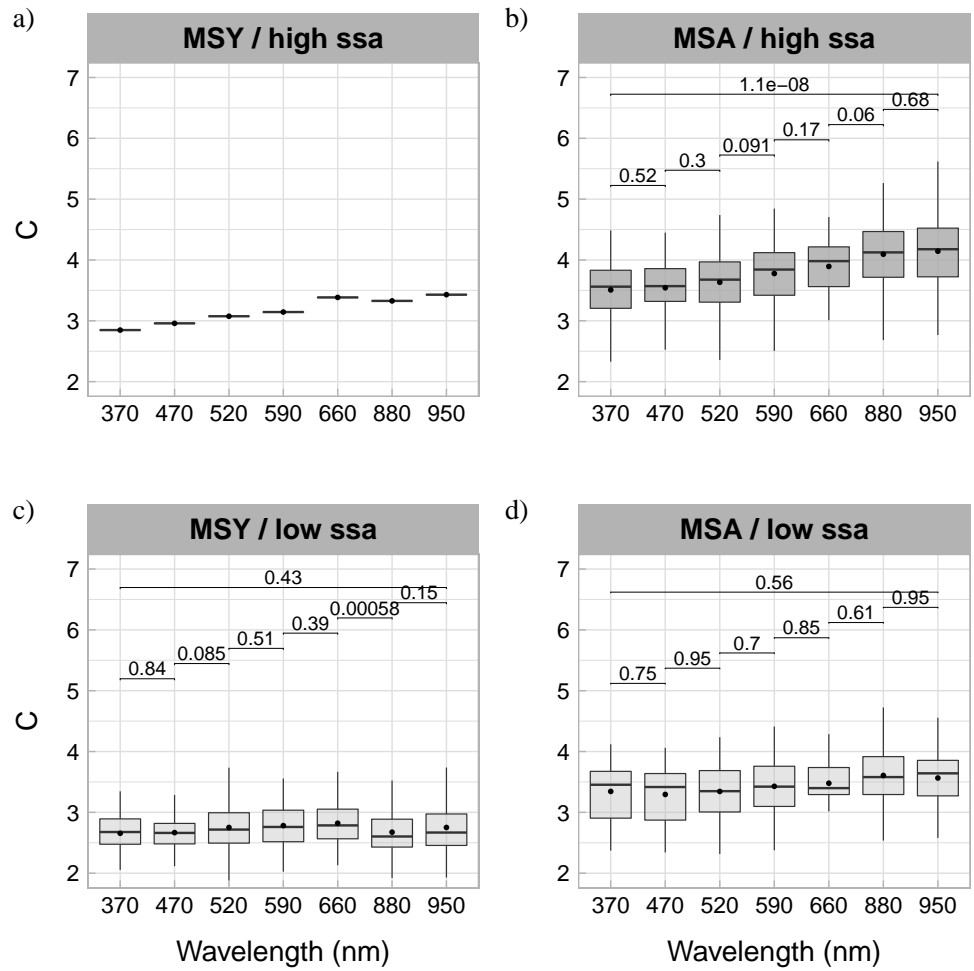

**Figure 5.** Wavelength dependence of C at MSY (a,c) and MSA (b,d) obtained comparing $b_{atn}$ from the AE33 measured at each wavelength and $b_{abs}$ inter/extrapolated to the same wavelength from the PP_UniMI. Box plots have been obtained as in Fig. 3 and separated into two categories depending whether the SSA was above (high ssa, upper panel) or below (low ssa, lower panel) the threshold at which C sharply increases. As in Fig. 3, the values above the box plots between adjacent wavelengths and between 370 and 950 nm wavelength box plots show the obtained p-values, with p<0.05 meaning a statistically significance difference.

wavelength dependence for high SSA at MSY. Despite this, a 17% increase of C with the wavelength, from 2.85 at 370 nm to 3.43 at 950 nm, for this single point was observed (cf. Fig. 5a). At MSY, similarly to MSA, the C did not show any dependence with the wavelength for SSA<0.95 (cf. Fig.5c). Thus, this analysis demonstrated that high SSA of the particles deposited on the filter tape can increase the C values, influencing at the same time its wavelength dependence.

We have shown in Section 3.1 that the sharp increase of C at high SSA at the stations herein analyzed can be associated 450 with the presence of particles dominated by dust, characterized by low SSAAE and BF and high AAE and SSA (Figs. S3,

S4 and S5). Therefore, we performed a similar C spectral dependence analysis as in Fig. 5, but separating the days affected by Saharan dust (dust) and the days without dust influence (no-dust). As shown in Fig. 6, no spectral dependence of C was observed during either dust and no-dust scenarios at MSY. This lack of dependence with the dust intrusions could be due to the limited number of off-line samples at MSY characterized by high SSA (1 out of 126). Thus, due to the low temporal

resolution of off-line PP_UniMI measurements, even during Saharan dust days the SSA at MSY rarely increased above the SSA threshold. Nevertheless, using high-time resolution data (cf. Fig. 1) the potential effect of dust particles to increase the SSA (and consequently the C) was evident at both MSY and MSA. At MSA (cf. Fig. 6) the C showed a statistical significant increase with wavelength for both dust and no-dust samples due to the fact the the samples with high SSA at MSA (86%) were well distributed between the two scenarios. Thus, these results confirmed that the SSA was the main parameter that influenced

the spectral behaviour of the C parameter.

To further explore the possible causes that contributed to the different C spectral dependencies observed, we performed a similar analysis as in Virkkula et al. (2015) by comparing the C and its wavelength dependence with different aerosol particles intensive optical properties, namely: SSA, BF and SSAAE. Virkkula et al. (2015) and Drinovec et al. (2017) have shown that the AE33 factor loading parameter, k, increases with increasing BF (smaller particles) and decreases with increasing SSA and

465 that the wavelength dependence of k also depends on these two optical properties as well as on the particle mixing state. In Fig. S9 we present a similar analysis by studying the effects of these intensive optical properties on the multiple scattering parameter C instead of k. Fig. S9 shows the slope of C with the wavelength (i.e. the wavelength-dependence of C) with SSA, BF, and SSAAE at the three sites. No clear relationship was observed between the C slope and the three intensive optical properties at both BCN and MSY. Moreover, the C slope at these two sites were close to zero for the considered intensive optical properties.

The observed lack of C gradient was again likely due to the fact that at BCN and MSY the SSA did not exceed the threshold value, even when the SSAAE indicated the possible presence of Saharan dust intrusions at MSY (cf. Fig. S9h). However, Fig. S9c shows that at MSA there was a shift of the C slope toward large positive values when SSA was above 0.95. Below this SSA threshold value, the C slope was close to zero confirming the reduced C wavelength dependence for low SSA values at MSA. Moreover, when the SSAAE(BF) at MSA (cf. Fig. S9i and S9f) decreased towards negative(low) values (Saharan

dust intrusions), the slope of the C increased, again confirming the potential of coarse Saharan dust to increase the SSA and, consequently, the C especially at the remote site. Note that, as already commented (cf. Fig. 6), the C slope kept positive values at MSA also for the samples not dominated by dust (SSAAE>0), thus further indicating the predominance effect of SSA on the C wavelength dependence. Thus, the results presented in Fig. S9 confirmed the effects of SSA on the C presented in Fig. 5 and 6.

The lack of points for BCN (none) and MSY (1 of 126) for large SSA values, specially above the SSA threshold obtained in Fig. 1, prevented from extrapolating the results to other measurement background conditions and further studies should be performed to better characterize the spectral behaviour of C and its dependency with the cross-sensitivity to scattering under different atmospheric conditions/scenarios. This is specially important, as already commented, in view of the contradictory results reported in literature (e.g. Weingartner et al., 2003; Segura et al., 2014; Bernardoni et al., 2020). The results presented

here clearly indicated that when the SSA exceeded a given site-dependent threshold, as determined using the method in Sect.

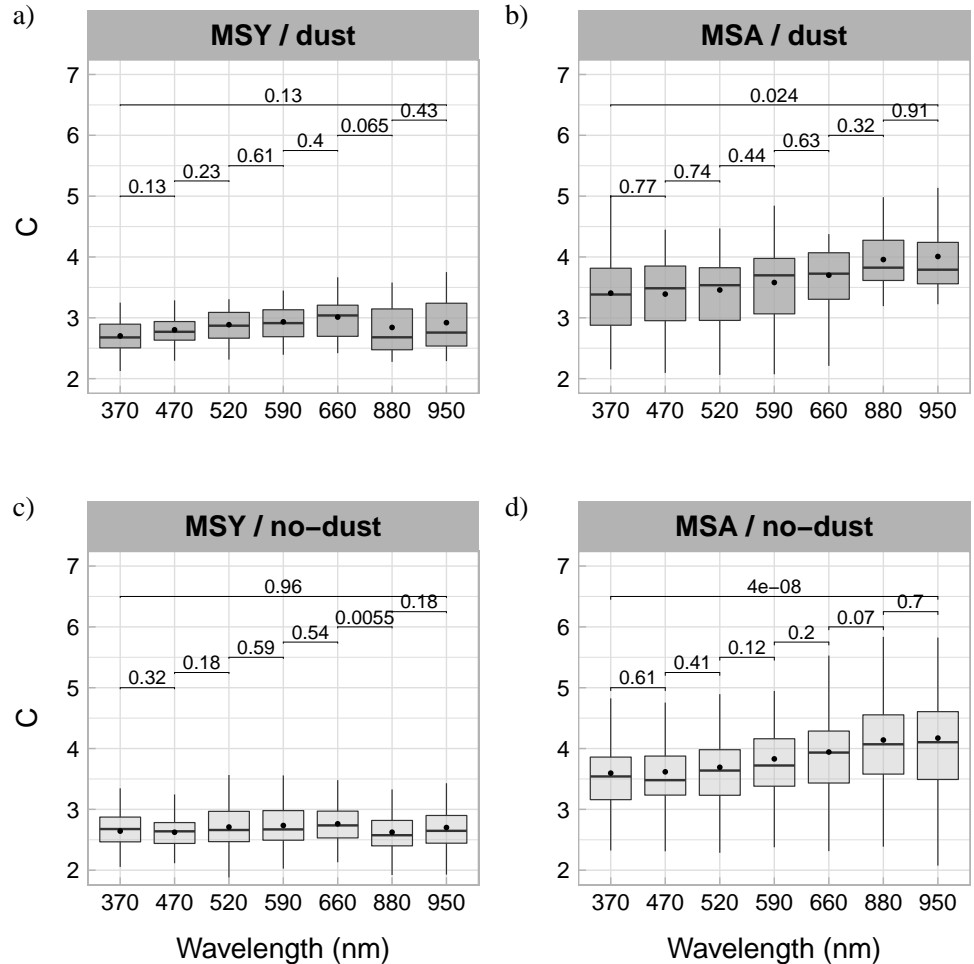

**Figure 6.** Wavelength dependence of C at MSY (a,c) and MSA (b,d) obtained comparing $b_{atn}$ from the AE33 measured at each wavelength and $b_{abs}$ inter/extrapolated to the same wavelength from the PP_UniMI. Box plots have been obtained as in Fig. 3 and separated into two categories depending whether Saharan dust outbreaks took place (dust) or not (no-dust). As in Fig. 3, the values above the box plots between adjacent wavelengths and between 370 and 950 nm wavelength box plots show the obtained p-values, with p<0.05 meaning a statistically significance difference.

3.1, the C values and its wavelength dependence increased. For the measurement sites considered here, Saharan dust outbreaks were identified as possible cause for SSA values higher than the threshold. However, from a general point of view, other factors, including the location of the measurement stations and/or absence of anthropogenic pollution, can determine the presence of a particle mixture with high or very high SSA.

Finally, we performed a sensitivity study on the effects that using a wavelength-dependent C (C($\lambda$)) had on the AAE derived from AE33 measurements, comparing the results with those obtained using the usual approach based on the application of a

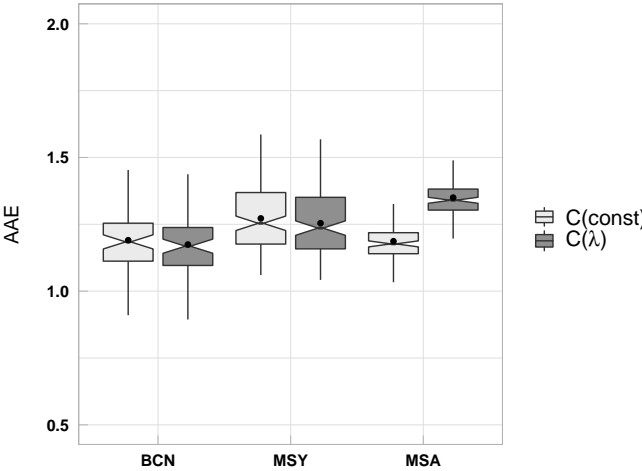

**Figure 7.** The absorption Ångström exponent (AAE) calculated with a constant C(const) and the wavelength-dependent C($\lambda$) for all stations. Box plots have been obtained as in Figs. 3 and 4, with the markers indicating the mean AAE values.

constant C factor (C(const)). Figure 7 shows that the AAE values for BCN and MSY did not present any significant variation (cf. Table S4), with AAE mean values of 1.19 $\pm$ 0.15 and 1.27 $\pm$ 0.12 (at BCN and MSY, respectively) for C(const), and 1.17 $\pm$ 0.15 and 1.25 $\pm$ 0.12 (for BCN and MSY, respectively) for C($\lambda$). These results for BCN and MSY were coherent with the
observed lack of spectral dependence of C at these two stations (Fig. 4). However, at MSA the observed increase of the C with the wavelength, introduced an increase of the AAE of around 13%, from 1.19 $\pm$ 0.07 (C(const)) to 1.35 $\pm$ 0.07 (C($\lambda$)). Similarly, Fig. S10a,b present the results of a sensitivity analysis performed to understand the effects that using a constant or a wavelength-dependent C had on the SSA at 470, 660 and 950 nm. As for the AAE, Fig. S10c shows no significant variation of SSA at the 3 considered wavelengths at BCN and MSY, again consistent with the observed lack of dependence of the C factor
with wavelength at these two sites. However, Fig. S10 shows a statistically significant increase of the SSA at MSA station of around 1.3% at 660 nm and 2% at 950 nm when using C($\lambda$) instead of C(const). Conversely, as expected, no statistically significant change was appreciated at the lower wavelength, 470 nm. This variation introduced by C($\lambda$) on AAE and SSA, although not large, is relevant since it occurs at the threshold of SSA value for which a substantial increase of the C as a function of SSA was observed, as shown in Section 3.1.

**4 Conclusions**

In this work we studied the multiple scattering parameter C for two filter tapes used in AE33 dual-spot aethalometers, i.e. the previously used M8020 and the currently used M8060 filter tapes. For this, we used data collected at three different background stations in NE Spain: an urban background station in Barcelona, BCN, a regional background station at Montseny,

MSY, and a mountain-top station at Montsec d'Ares, MSA. We obtained the C correction factor comparing the AE33 atten-
uation measurements with the absorption coefficients measured from MAAP instruments, and used simultaneous scattering
measurements from an integrating nephelometer to characterize the cross-sensitivity to scattering of C. Moreover, we studied
the C wavelength dependence at the three sites comparing the AE33 attenuation coefficients with the absorption coefficients
from the off-line multi-wavelength PP_UniMI .

We presented here a novel approach to characterize the cross-sensitivity to scattering of the C correction factor. This approach
consisted in fitting the measurements of the C versus SSA. The fits provided the constant $C_f$ and a cross-sensitivity factor $m_S$.
We applied the fits to the M8020 filter tape at MSY and MSA, and we obtained higher cross-sensitivity values of the C factor
($1.8\pm0.1\%$ and $3.4\pm0.1\%$ at MSY and MSA, respectively) compared to those reported in the literature (around 1-1.5 %). For
the first time, we characterized here the cross-sensitivity to scattering also of the new M8060 filter tape. We obtained a higher
cross-sensitivity to scattering for the M8060 than for the M8020 filter tape, with values of $1.6\pm0.3\%$, $3.0\pm0.1\%$ and $4.9\pm0.1\%$
for BCN, MSY and MSA, respectively. The multiple scattering parameter, $C_f$, for the M8020 filter tape was $2.21\pm0.01$ at
MSY and $1.96\pm0.02$ at MSA. For the M8060 filter tape the fit led to $C_f$ values of $2.50\pm0.02$ at BCN, $1.96\pm0.01$ at MSY,
and $1.82\pm0.02$ at MSA. The consequence of this cross-sensitivity to scattering resulted in a large increase of the C values,
up to 3-fold increase, for SSA values above 0.9-0.95. This significant increase of the C factor at high SSA, if not accounted
for, can lead to a large overestimation of both eBC concentrations and absorption coefficients measured by aethalometers.
This can be especially relevant at sites typically characterized by an aerosol mixture with high SSA. In fact, the effect of this
cross-sensitivity to scattering of C was the likely reason explaining the higher C values reported in literature for mountain-top
and Arctic measurement stations. Here, we observed larger C values and higher cross-sensitivity to scattering at the mountain
station and much less C variability at the urban site, where the SSA rarely exceeded the SSA threshold from which changes in
C can be observed.

Overall, the main difference between the two filter tapes studied here was the higher cross-sensitivity to scattering observed
for the currently used M8060 filter tape compared to the previously used M8020 filter tape. Despite the different cross sensi-
tivity to scattering, both filter tapes showed average C values which fall within the measurement uncertainties.

We found an average multiple scattering parameter C at 637 nm of 2.29, 2.29, 2.36 for the M8020 filter tape and of 2.44, 2.23
and 2.51 for the M8060 filter tape, for BCN, MSY and MSA measurement stations, respectively. Due to the dominant effect
of SSA on the C, the obtained C factors showed seasonal and diel variability at the three sites that mirrored the variability
of SSA. At MSY and MSA higher C values were on average observed in summer due to changes in the physical-chemical
aerosol properties that led to SSA values on average higher in summer than in winter. A larger fraction of dust particles and
formation of secondary organic aerosols and secondary sulfates likely explained the observed increase of C in summer at these
regional/remote sites. However, at the urban background station of BCN the C values remain fairly constant throughout the
year.

We also analyzed the wavelength dependence of the C parameter for the M8060 filter tape at BCN, MSY and MSA by
comparing the AE33 attenuation data with the off-line PP_UniMI absorption measurements performed on selected MAAP
spots. Overall, we found a statistically significant increase with the wavelength, from 3.47 for 370 nm to 4.03 for 950 nm at

the mountain-top station (MSA), whereas at BCN and MSY background stations no statistically significant dependence was found. The reason for the lack of wavelength dependence of the C at BCN and MSY was the lack of MAAP spots characterized by high SSA. Thus, due to the low temporal resolution of off-line PP_UniMI measurements, the SSA at MSY and, especially, at BCN rarely increased above the SSA threshold. Conversely, the wavelength-dependence of C at the mountain station was due to the high probability of measuring SSA values higher than the site-dependent SSA threshold, from which the C values start to increase. For this analysis, we studied the C wavelength dependence separately for samples characterized by high SSA (higher than the site-dependent threshold) and low SSA and observed that at MSA no dependence of the C with the wavelength was observed for samples with low SSA, whereas a clear dependence was observed for the sample with high SSA. Thus, the analysis presented here demonstrated that high SSA of the particles deposited on the filter tape can increase the C values influencing at the same time its wavelength dependence. Interestingly, only one sample (out of 126) collected at MSY regional station was characterized by high SSA and for this sample the calculated C strongly increased with wavelength. The results presented here clearly indicated that when the SSA exceeded a given site-dependent threshold, the C values and its wavelength dependence increased. For the measurement sites considered here, Saharan dust outbreaks were identified as possible cause for SSA values higher than the threshold. However, other factors,including the location of the measurement stations and/or the absence of anthropogenic pollution, can determine the presence of a particle mixture with high or very high SSA. We also investigated the effect of considering a wavelength-dependent C at MSA station compared to using a constant C on the absorption Ångström exponent (AAE) and the single scattering albedo (SSA) through sensitivity tests. Results revealed an increase of the AAE by 13% and an increase of the SSA by 1.3% when using the wavelength-dependent C factor compared to using a constant C factor (i.e. with no $\lambda$–dependence). This effect may impact any source apportionment method which takes into account the multi-wavelength absorption values from the AE33 (e.g. the Aethalometer model).

In summary, based on the results herein presented, the absorption coefficients from AE33 data can be corrected with different degrees of confidence depending on the information available to estimate the multiple scattering parameter C:

– A tailored dynamic multiple scattering parameter can be obtained if on-line simultaneous reference absorption measurements are available. In this case, a dynamic C with high temporal resolution can be obtained, allowing an in-situ correction of AE33 data and allowing studying, for example, diel/seasonal cycles of the multiple scattering parameter. Here we used on-line MAAP absorption measurements at one wavelength for the determination of a dynamic C at the same MAAP wavelength.

– If independent reference multi-wavelengths absorption measurements are available, then the dependence of the multiple scattering parameter with wavelengths can be studied. Here we determined the wavelength dependence of the multiple scattering parameter C by using the polar photometer (PP_UniMI) off-line absorption measurements performed on the MAAP filter spots and by comparing the off-line PP_UniMI measurements with AE33 attenuation data integrated over the MAAP filter spots time stamp.

– If reference absorption measurements are not available for the experimental determination of the C, then the average values of the multiple scattering parameter provided here for three different measurement stations can be used as reference.

- If both independent reference absorption measurements and scattering measurements are available, then the cross sensitivity to scattering of the multiple scattering parameter C can be determined by studying the relationship between C and single scattering albedo (SSA). In this case, a parameterization can be obtained relating C and SSA.

- If SSA measurements are not available, this work provides parameterized formulas that allow calculating C over a wide range of SSA values.

Finally, the C values obtained in this work for different station types (urban, regional, remote) may serve as reference for similar background measurement sites where the methodology presented here cannot be applied. Nevertheless, discrepancies may arise due to the possible differences in aerosol sources, composition and mixing state at different sites that, accordingly, will results in different aerosol particle optical properties. Similar analysis performed at other measurement sites with similar features may reduce the uncertainties around the applicability of the results presented here to other stations.

## Appendix A:  Absorption coefficient relationship between a MAAP and a PP_UniMI polar photometer for MSA station

This appendix aims to show the result of applying the same methodology as in Section 3.1 of Valentini et al. (2020b) to the PP_UniMI analyzed dataset for obtaining the bias for the MSA station in the absorption coefficient measurements between the MAAP and the PP_UniMI polar photometer A1. It consists on the application of a Deming regression fit, which results in a slope of $0.81 \pm 0.01$ for our dataset.

*Code and data availability.*  The Montseny and Montsec data sets used for this publication are accessible online on the WDCA (World Data Centre for Aerosols) web page: http://ebas.nilu.no. The Barcelona data sets were collected within different national and regional projects and/or agreements and are available upon request. The code used for analysis can be obtained upon request to the corresponding author.

*Author contributions.*  DC, SV, RV and VB performed and analyzed the measurements with the PP_UniMI polar photometer. NP, CR, MP, AA and JYD carried out the maintenance and supervision of the BCN, MSY and MSA supersites. AA, GM, MP and XQ played a crucial role in the processes of shaping the manuscript structure as well as helping with the data analysis. JYD developed the data process, the analysis of the results, and summarized and expressed them in this manuscript. All authors provided advice regarding the manuscript structure and content as well as contributed to the writing of the final manuscript.

*Competing interests.*  At the time of the research, MR and MI were also employed by the manufacturer of the Aethalometer AE33.

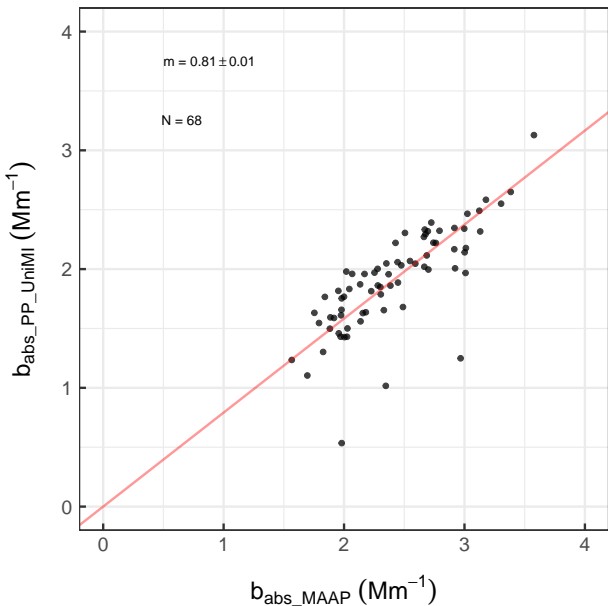

**Figure A1.** Comparison between the aerosol absorption coefficient measured by PP_UniMI on sample spots ($\sigma_{ap\_PP\_UniMI}$) and the MAAP photometer ($\sigma_{ap\_MAAP}$).

*Acknowledgements.* Measurements at Spanish sites (Barcelona, Montseny and Montsec d'Ares) were supported by the Spanish Ministry of Economy, Industry and Competitiveness and I+D+I "Retos Colaboración" funds under the CAIAC project (PID2019-108990PB-100), by the Generalitat de Catalunya (AGAUR 2017 SGR41 and the DGQA) and the European Commission via ACTRIS-IMP (project 871115). We acknowledge support of the COST Action CA16109 COLOSSAL. GM acknowledges support from the Slovenian Research Agency program P1-0385 "Remote sensing of atmospheric properties". IDAEA-CSIC is a Centre of Excellence Severo Ochoa (Spanish Ministry of Science and Innovation, Project CEX2018-000794-S).

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
