# Peer review of "Determination of the multiple-scattering correction factor and its cross-sensitivity to scattering and wavelength dependence for different AE33 Aethalometer filter tapes: A multi-instrumental approach"

_Atmospheric Measurement Techniques, 2021_

## Author Comment (AC1)

*Referee comment on "Determination of the multiple-scattering correction factor and its cross-sensitivity to scattering and wavelength dependence for different AE33 Aethalometer filter tapes: A multi-instrumental approach" by Jesús Yus-Díez et al., Atmos. Meas. Tech. Discuss., https://doi.org/10.5194/amt-2021-46-RC3, 2021*

**Answer from the authors to referee #1**

On behalf of all the authors of the manuscript, we would like to acknowledge the work done in the review as well as the suggestions and comments for improving the study.

Hereafter we will answer and resolve the comments. Any minor comment, typo or writing corrections will be directly corrected in the manuscript.

**Please check second Ramanathan's reference. It is shown as published in 2020, when it was actually published in 2001.**

Checked and corrected.

---

## Author Comment (AC2)

*Referee comment on "Determination of the multiple-scattering correction factor and its cross-sensitivity to scattering and wavelength dependence for different AE33 Aethalometer filter tapes: A multi-instrumental approach" by Jesús Yus-Díez et al., Atmos. Meas. Tech. Discuss., https://doi.org/10.5194/amt-2021-46-RC2, 2021*

**Answer from the authors to referee #2**

We thank the reviewer #2 for her/his very useful comments that contributed to improve the quality of this manuscript. Thanks to the reviewer comments some results are now better described and presented.

One of the most important changes in the new version of the manuscript raised from the reviewer comment was about the need to study the C wavelength dependence for samples characterized by SSAAE<0 (dust) and for samples with SSAAE>0 (no-dust). This sensitivity test, suggested by the reviewer, was useful to better understand the role of dust particles in the observed spectral dependence of the C.

The results of this analysis (Fig. 5, 6) revealed that at MSA the spectral dependence of the C was statistically significant for both sample categories (dust, no-dust). To further explore the reasons for this behavior, we performed a similar analysis by studying the spectral dependence of the C for high SSA and for low SSA. The SSA threshold at each station was determined based on the results presented in Fig. 1 and was selected as the SSA value from which the C starts to increase. The results of this additional sensitivity study revealed that when the SSA was lower than the SSA threshold, then the spectral dependence of the C was not s.s. at both MSA and MSY. Conversely, for the samples characterized by SSA higher than the threshold we observed a s.s. increase of C with wavelength at both MSA and MSY.

Thus, high SSA values can both increase the C values and make the C dependent on the wavelength. We, however, confirm, in the revised version of the manuscript, the fact that Saharan dust can contribute to make the C dependent on the wavelength. Because, as shown in Figure 1 and in previous published papers, Saharan dust tends to increase the SSA (and consequently the C) at the measurement stations considered here.

The main conclusion from this analysis is that at all sites were high or very high SSA can be expected, then the C will increase and will be depend on the wavelength.

Another important improvement of the revised version of the manuscript was that, as suggested by the reviewer, we added at the end of the Conclusion section a short summary where we suggest which could be the best strategy to correct AE33 data depending on the measurement available.

For simplicity, we report below the new Abstract:

[revised manuscript text omitted]

Hereafter we will answer and resolve the comments. Any minor comment, typo or writing corrections will be directly corrected in the manuscript.

**Main comments**

**1. In the methodology part, four different C constant are described (instr, lambda, f and eff). Moreover a new Cconst is introduced at page 16. The first point is that the introduced abbreviation are sometimes "rephrased" or misused (e.g. C(C(λ)) or "the wavelength-dependent C"). But the most important point is that most of the result section just mention C without any specification. This can lead to misunderstanding and does not help the reader to easily understand this technical study.**

We thank the reviewer for noticing the lack of consistency. We checked the nomenclature of C simplifying it so that it is clearer for the reader.

We replaced the term $C_{eff}$ with C throughout the manuscript, since the use of $C_{eff}$ was redundant. Also, we changed the term $C_{instr}$ with $C_0$ for abbreviation purposes. Concerning $C_{const}$, we changed the legend of the corresponding figure (Fig. 7) and corresponding text by referring to it as C(const). Moreover, for any variable, x, which depended on the wavelength we used the syntax, such as in equations 4 and 5, we used the syntax $x(\lambda)$.

**2. The influence of Saharan dust outbreaks on the Ceff values and on its wavelength dependence is clearly demonstrated in sections 3.2 and 3.3. Section 3.1 already introduces and discuss the dust influence but without real proofs. I think that the structure of the paper should be reviewed (to some extent) so that misinterpretations of results of section 3.1 are removed (see the 11 comments I had on section 3.1 during the reading). This can perhaps be solved by a result section followed by a discussion section or by another ordering of the results description.**

We thank the reviewer for this comment on the overall structure of the paper. We have rearranged the order of the results section. Now, in subsection 3.1 (i.e. the previous subsection 3.3) we present the cross-sensitivity of C to scattering and comment about the importance of African dust in increasing the SSA (and consequently the C values); the new subsection 3.2 (i.e. the previous subsection 3.1) presents the average values of the C and the seasonal variations of C and SSA; the new subsection 3.3 (i.e. the previous subsection 3.2) presents the wavelength dependence of the C.

Given the extension of the subsections, we do not present the aforementioned changes in this document. All the changes can be found in the revised manuscript and in the manuscript with all the changes tracked and marked.

Moreover (see also the reply to Reviewer comment #3), we summarized at the end of the conclusion section the procedure to correct the AE33 data depending on the information available (e.g. independent absorption and SSA measurements).

**3. You had access to a reference instruments to study the behavior of the multiple scattering constant at three stations with different aerosol composition. A section describing the best way to correct AE33 for the multiple scattering constant as a function of e.g. the type of aerosol, the SSA, the presence of dust as well as if scattering coefficient or MAAP data are available would help the reader and increase the value of your paper.**

As suggested by the referee, we included a brief summary with the main remarks at the end of the conclusions in order to summarize how to correct the AE33 data depending on the information available

"In summary, based on the results herein presented, the absorption coefficients from AE33 data

can be corrected with different degrees of confidence depending on the information available to estimate the multiple scattering parameter C:

- A tailored dynamic multiple scattering parameter can be obtained if on-line simultaneous reference absorption measurements are available. In this case, a dynamic C with high temporal resolution can be obtained, allowing an in-situ correction of AE33 data and allowing studying for example diel/seasonal cycles of the multiple scattering parameter. Here we used on-line MAAP absorption measurements at one wavelength for the determination of a dynamic C at the same MAAP wavelength.
- If independent reference multi-wavelengths absorption measurements are available, then the dependence of the multiple scattering parameter with wavelengths can be studied. Here we determined the wavelength dependence of the multiple scattering parameter by using the polar photometer (PP_UniMI) off-line absorption measurements performed on the MAAP filter spots and by comparing the off-line PP_UniMI measurements with AE33 attenuation data integrated over the MAAP filter spots time stamp.
- If reference absorption measurements are not available for the experimental determination of the C, then the average values of the multiple scattering parameter provided here for three different measurement stations can be used as reference.
- If both independent reference absorption measurements and scattering measurements are available, then the cross sensitivity to scattering of AE33 data can be determined by studying the relationship between C and single scattering albedo (SSA). In this case, a parameterization can be obtained relating C and SSA.
- If SSA measurements are not available, this work provides parameterized formulas that allow calculating C over a wide range of SSA values."

**4. The impact of the two different filter tapes should be also summarized anywhere.**

We added the following sentences to summarize the differences between the two filter tapes and their impact:

After equation 4 (lines 217-220):

"Note that the new filter tape M8060 structurally differs from the old filter tape M8020 in filter fibers material, thickness and density, thus leading to different $C_0$ values (details can be found in the following online document from Magee Scientific: https://mageesci.com/tape/Magee_Scientific_Filter_Aethalometer_AE_Tape_Replacement_discussion.pdf)."

We have included in lines 526-529 a few lines commenting on the difference between filter tapes M8020 and M8060.

"Overall, the main difference between the two filter tapes studied here was the higher cross-sensitivity to scattering observed for the currently used M8060 filter tape compared to the previously used M8020 filter tape. Despite the different cross sensitivity to scattering, both filter tapes showed average C values which fall within the measurement uncertainties."

**5. Finally, a lot of minor points such as 1) English language, 2) coherence of parameter's abbreviations between the figures and the text, 3) repetitions in the results description, 4) right citations (i.e. cite the right paper in the right context) should had been corrected before the first submission.**

Following the reviewer suggestion, we have tried to improve each of these minor points highlighted by the reviewer.

**Minor comments**

**1. 1 line 10: C(λ) is not only the parameter with the greatest uncertainty but above all the parameter with the greatest impact on the determination of the absorption coefficient.**

We have added the following statement to the line to make it clearer. Lines 8-10:

"The multiple-scattering correction factor (C), which depends on the filter tape used and on the optical properties of the collected particles, is the parameter that showed both the greatest uncertainty and the greatest impact on the absorption coefficients derived from the AE33 measurements."

**2. line35-37: I'm not sure that no standard aerosol particles are available for instrument calibration. Polystyrene balls are available to calibration scattering measurement. For absorption, some recent developments are also available (e.g. https://www.tandfonline.com/doi/pdf/10.1080/02786826.2018.1536818?needAccess =true). Please verify and if necessary correct this sentence.**

Indeed, scattering measurements do have a standard calibration procedure, and as stated by Ess and Vasilatou (2019), flame-generated soot is increasingly being used in academia and industry as engine exhaust soot surrogate for atmospheric studies and instrument calibration. We have removed the sentence from the line 35-27 since it was wrong and we added a new sentence where we comment about the recent developments in the context of the absorption measurements calibration.

Lines 58-60:

"Moreover, there is also the need of standard aerosol particles to use as reference for quality assurance of absorption measurements such as the recently developed flame-generated soot (Ess and Vasilatou, 2019).

**3. line38-39: this is already the case in part of the Europe and in north America (https://acp.copernicus.org/articles/20/8867/2020/)**

We modified the sentence, in lines 43-45 as follows:

"However, this influence is likely to be reduced over the coming decades as air pollution measures are implemented around the world (Samset et al., 2018), as it is already the case in parts of Europe and North America (Collaud Coen et al., 2020)."

**4. Measurement site: a reduction of the number of cited references could really help the reader to gain time in searching information related to the stations and the aerosol sources.**

We have reduced the number of cites per site as suggested. The new sentence is reported below:

"A detailed characterization of the three measurement stations can be found in previous works (e.g. Querol et al. (2001); Rodrıguez et al. (2001); Reche et al. (2011) for BCN; Pérez et al. (2008); Pey et al. (2009); Pandolfi et al. (2011) for MSY; Pandolfi et al. (2014a); Ripoll et al. (2014); Ealo et al. (2016) for MSA)."

**5. Line 195, replace ";" by "," after (2014)**

Corrected as suggested by the referee.

**6. Line 238: Schmid et al (2006) did not report a correction including directly the scattering coefficient. He used the SSA value as equ (8). Contrarily to Schmid, Arnott substracted a fraction of the scattering from bATN before to divide by β (corresponding to the multiple scattering constant). Segura took the Schmid algorithm. Please clarify**

Given that we followed a similar approach as the one carried out by Arnott et al. (2005), we have modified the corresponding part of the manuscript to avoid confusion:

Lines 255-256:

"Taking this dependence into account and following a similar approach as Arnott et al. (2005), and Segura et al. (2014), we parameterized the light attenuation coefficient as:"

And also, lines 262-264:

"If we rearrange eq. (7) by expressing the scattering coefficient through the single scattering albedo, we obtain the dependence of the absorption as a function of SSA (eq. 8), similarly to eq. 17 in Schmid et al. (2006)."

**7. Line 241: introduce f(ATN) directly in equation 4 for clarity purpose.**

Following the reviewer suggestion, we included f(ATN, λ) in equation 4, which is in line 215 as:

$$b_{atn}(\lambda) = eBC(\lambda) \cdot \sigma_{abs}(\lambda) \cdot C_0 = \frac{S}{F} \frac{\Delta ATN(\lambda)}{\Delta t} \cdot f(ATN, \lambda)$$

**8. Line 242: using AE33 the filter loading correction is performed by the instrument. Why is it mention here? Please give your own argumentation/measurement to say that f(ATN) can be assumed to be close to 1 since your correction without loading correction is not similar to Schmid one.**

Indeed, the expression we used is not correct. In fact, we did not assume in our analysis that f(ATN) is close to 1. Since the filter loading correction is performed by the instrument and the f(ATN) function is already taken into account by the AE33 software, we have accordingly modified the sentence as follows:

Line 241 has been moved to lines 216-217 as follows:

"where f(ATN,λ) is the function which contains all the corrections, i.e. filter loading and leakage, which are performed by the AE33 for each wavelength (Drinovec et al., 2015)."

**9. Line 253: which other (than scattering) cross-sensitivity could be taken into account? The sentence line 251-253 would be clearer if reformulated.**

Indeed, we only refer to the cross-sensitivity to scattering in this work. Another possible cross-sensitivity could be the cross-sensitivity to particle size distribution, as smaller particles can

penetrate more deeply in the filter matrix then coarse particles. However, the study of other cross-sensitivities is far from the scope of our work and we prefer not to mention it in the manuscript because we cannot demonstrate it with the available measurements.

We have modified these sentences, now in lines 267-270 as follows:

"The effective multiple scattering parameter, C, depends on the physical properties of collected particles. By comparing data from different instruments (AE33, MAAP, and nephelometer) we were able to parameterize the cross-sensitivity of the C to scattering (Eq. 8). Eq. 9 shows that the actual AE33 cross-sensitivity to scattering is more pronounced when the measured aerosol particles have higher SSA, whereas for particles with lower SSA Eq. 9 converges to eq. 5."

**10. Lines 256 and 257, 260 (please check the whole document): for your own correction, you defined up to now Cinst, Clambda, Cf and Ceff. Please use one of these instead of "C".**

We have checked and modified accordingly the whole document so that now there is consistency when we refer to the multiple scattering parameter through the manuscript.

**11. Lines 257-259: Similarly to scattering coefficient, absorption coefficient can also depends on the particle size and mixing state. Nothing is wrong and it is good to study the dependency on the shape, size and mixing state. Anyhow this is not scientifically and grammatically precise and it should be reformulated.**

We thank the reviewer for the comment on the paragraph. We have modified it so that it is now clearer. Now it corresponds to the range of lines 271-274.

"By analyzing the dependency of the effective multiple scattering parameter C with the SSA we obtained the experimental fit constants ($C_f$ and $m_s$) that describe the relationship between C and SSA. Furthermore, we will present in Section 3.1 how the cross-sensitivity to scattering of C depended on some intensive aerosol particle optical properties that strongly depend on aerosol particles size distribution and chemical composition (Figs. S3-S5)."

Also, we have included the following line in Sect. 3.1. at lines 348-350:

"The observed dependency on aerosol particle intensive optical properties demonstrated that both particle size distribution and chemical composition can affect the reported C vs. SSA relationships."

**12. Lines 260-270: please mention the measuring time granularity of each instrument.**

The time granularity of each instrument varied during the measurement period and ranged from 1 min, when data were collected with the NOAA CPD3 acquisition software or other logging systems, to 5 min when Aurora 3000 integrating nephelometer data were logged with the corporative Aurora logging software.

We have added the following sentence in the manuscript at lines 285-286:

"Moreover, the time granularity of the measurements varied between 1 to 5 minutes, depending on the software used for data logging (see Table S1)."

Table S1: Measurement time granularity for the AE33, MAAP and nephelometer at BCN, MSY

and MSA measurement stations.

| INSTRUMENT | STATION | TIMESTAMP (PERIOD) |
|---|---|---|
| AE33 | BCN | 1 min |
| | MSY | 1 min |
| | MSA | 1 min |
| MAAP | BCN | 1 min |
| | MSY | 1 min |
| | MSA | 1 min |
| NEPHELOMETER | BCN | 1 min |
| | MSY | 5 min (2013-February February 2017); 1 min (February 2017-2020) |
| | MSA | 5 min (2013-February February 2017); 1 min (February 2017-2020) |

**13. Line 274: the scattering coefficient was probably also similarly averaged?**

Indeed, we have modified the sentence so that it also indicates the averaging procedure applied to the scattering.

Lines 292-294:

"Since the off-line PP_UniMI measurements were performed on the MAAP spots, the measured attenuation and scattering coefficients from AE33 and nephelometer, respectively, were averaged over the timestamp of each one of the selected MAAP spots."

**14. Line 288-292: this is partly the same information. Please reformulate these sentences. I also don't understand "In the latter case" which seems to relate to the Deming regression. So you used two methods, the Deming including the uncertainties of the instruments and another one using the statistical density distribution of the C factor?**

We have reformulated these sentences as suggested by the reviewer.

Indeed, we have used two methods to obtain the C factor since both can be found in the literature, e.g.: Backman et al. (2017) and Bernardoni et al. (2020).

"In the latter case" refers to the method using the statistical density distribution of C.

We have modified this section to clarify this point. Due to the rearranging of the sections, it is now in the lines 352-358.

"Here we present the average values and the seasonal cycle of the C factor calculated at 637nm at BCN, MSY and MSA. We analyzed the multiple scattering parameter C values both through a Deming regression, taking into account the measurement error of the MAAP (12%; Petzold and Schönlinner, 2004) and of the AE33 (15%; Zanatta et al., 2016; Rigler et al., 2020), and by calculating the median value of the C factor density distribution. The uncertainties of the C factor were derived as either the methodological error from the regression slope of the Deming fit, or as the half-width at half maximum (HWHM) of the density distribution of the C factor. We present here the results from both the aforementioned methods because both methods have been reported in literature (e.g. Backman et al., 2017; Bernardoni et al., 2020; cf. Table 1 in this work)."

**15. Line 295: the median is done with all the simultaneous measurements of the three instruments? How is the uncertainties computed? There is a mixing between Deming method and averaging that should be removed. Please give the results after the description of the method and end with a comparison between them. Similarly you give the results first for M8060 and then for M8020 for the median and in inverse the order for the Deming method. This makes the comparison much more difficult.**

The median is done with all the measurements available of the AE33 instruments deployed at each station (Fig. S1). As mentioned in the paragraph between the lines 352-358, the uncertainties are computed as the half-width at half maximum (HWHM) of the density distribution.

We have reordered, as suggested, the paragraphs, so that it makes the comparison easier. It can be found at lines 352-375:

"Here we present the average values and the seasonal cycle of the C factor calculated at 637nm at BCN, MSY and MSA. We analyzed the multiple scattering parameter C values both through a Deming regression, taking into account the measurement error of the MAAP (12%; Petzold and Schönlinner, 2004) and of the AE33 (15%; Zanatta et al., 2016; Rigler et al., 2020), and by calculating the median value of the C factor density distribution. The uncertainties of the C factor were derived as either the methodological error from the regression slope of the Deming fit, or as the half-width at half maximum (HWHM) of the density distribution of the C factor. We present here the results from both the aforementioned methods because both methods have been reported in literature (e.g. Backman et al., 2017; Bernardoni et al., 2020; cf. Table 1 in this work).

The density distribution of the C factor obtained from the ratio (with a variable time resolution, as aforementioned in Sect. 2.3.1), showed a quasi-Gaussian distribution at the three measurement sites with a small tail toward higher C values (Fig. 2).

The median values of the C factor for the M8020 filter tape 2.29±0.48, 2.29±0.46, 2.36±0.59 for BCN, MSY and MSA, respectively. These values were on average similar or slightly lower (with differences less than a 7%) compared to the median C values obtained for the M8060 filter tape of 2.44± 0.57, 2.23±0.30, and 2.51±0.71 (cf. Table 1). The Deming regression fit results (Fig. S6) showed C values of 1.99±0.02, 2.05±0.02, and 2.21±0.03 (at BCN, MSY and MSA, respectively) for the M8020 which were slightly lower (with differences <10%) compared to the C values of 2.20±0.02, 2.13±0.01, and 2.05±0.02 obtained for the M8060. Note that the uncertainties from the Deming regression were lower compared to the uncertainties derived as HWHM of the distributions because the Deming regressions were performed using binned data (cf. Fig. S6). This also was the likely explanation for the lower C values on average obtained with the Deming regression compared to the median values of the density distribution. The difference of the C values between both methods ranged between 4-18% depending on the

filter tape/measurement station considered (cf. Table 1). However, both methods were consistent and provided higher C factor for the M8060 than for the M8020 filter tape.

As reported in Table 1, overall, higher C values were found at MSA, where both the SSA and the cross-sensitivity of the filter tape to scattering were higher compared to MSY and BCN (cf. Figs. 5 and S7). The C values for the AE33 M8020 and M8060 filter tapes obtained at urban background stations in Rome Rome (Valentini et al., 2020a) and Leipzig (Müller, 2015; Bernardoni et al., 2020) were in the same range as those found in this work for BCN (Table 1)."

**16. Lines 302-304: The median method leads to larger values that can reach up to 22% (M8060 for MSA) of the Deming method. This cannot be considered as marginally.**

We agree with the reviewer and for this reason the differences in the C values from the two methods were highlighted (see also reply to comment #15). We added/modified the following sentence in the lines 366-371:

"Note that the uncertainties from the Deming regression were lower compared to the uncertainties derived as HWHM of the distributions because the Deming regressions were performed using binned data (cf. Fig. S6). This also was the likely explanation for the lower C values on average obtained with the Deming regression compared to the median values of the density distribution. The difference of the C values between both methods ranged between 4-18% depending on the filter tape/measurement station considered (cf. Table 1). However, both methods were consistent and provided higher C factor for the M8060 than for the M8020 filter tape."

**17. Line 304: medians are precisely used to be not sensitive to extremes!! Figure 1: mention in the figure caption that the C (once again, which one?) comes from the medians. It could have been computed from Deming for e.g. each day of simultaneous measurements.**

We have included the following in the Figure caption of, now, Fig. 2.

"Density distribution of the C factor for each filter type, M8020 and M8060, as obtained through eq. (5) using both attenuation coefficient from the AE33 and the absorption coefficient from the MAAP."

**18. Lines 307-308: you didn't use the same data for both methods? If this is right, how can you compare the results?**

We have used the same data, however as can be seen in Fig. S7, we proceed to bin the data in the scatterplot for a better visualization. We applied the Deming regression to binned data to obtain a result comparable with Bernarndoni et al. (2020) as reported in Table 1 in the manuscript.

**19. What shall we trust/use, C from median or Deming? You have to discuss it and give your opinion about the best solution. If the Deming method allows detecting sensitivity to scattering by the value of the intercept, does it mean that you don't used Ceff that is corrected for the scattering? If you do not use Ceff, I don't understand which one you used and why you present this before the Ceff results. If Ceff is effectively used, why is there a remaining scattering sensitivity?**

We have used both methods, i.e. the Deming regression and the median of the density

distribution, to obtain the C factor since both methods can be found in the literature, e.g.: Backman et al. (2017) and Bernardoni et al. (2020) (cf. Table 1 of this manuscript).

We think that the most precise method for obtaining the cross-sensitivity to scattering is to apply the fit to eq. 9 using a scattering coefficient from independent measurements. It is a first approximation, since the cross-sensitivity to scattering may depend on other properties such as the particle size distribution. However, we expect that the fit is more precise than the intercept obtained through the Deming regression.

We did not correct the data for scattering when calculating the median of the C since our aim was to characterize the average values of the C at each station including the effects that differences in aerosol types caused on to the cross-sensitivity to scattering of the filter. The C values used for calculating the median were obtained by following eq. 5.

With the reorganization of the manuscript, we expect to have made clearer how the average values and the cross-sensitivity to scattering were obtained.

**20. Lines 318-322: Do you mean that both Saharan dust outbreaks and secondary organic aerosols/sulfate particles increase both the particle scattering efficiency and SSA? OK for the secondary particles, but dust particles absorb light and are usually/partly big particles. The impact of dust particles on SSA is then not straightforward.**

We have reformulated this part of the manuscript to avoid confusion.

Lines 385-389:

"Although dust particles can absorb radiation (e.g. Sokolik and Toon, 1999; Di Biagio et al., 2019), the effect of Saharan dust outbreaks at the measurement stations considered here was to increase the SSA over the average values. In fact, as shown by Pandolfi et al. (2014), both scattering and absorption increased at MSY and MSA during Saharan dust outbreaks, but the resulting SSA was higher compared to other atmospheric scenarios typical of the area under study."

**21. Lines 322-324: to my opinion, C is more constant at MSY than at BCN. The variability for each season is lower and the inter-season variability is quite similar.**

As observed by the reviewer, and as reported in Fig. 3 in the manuscript, the variability for each season of the C at BCN was higher than at MSY. This was likely due to a higher SSA variability for each season observed at the BCN station, as seen in Fig. S7 reported below. However, we think that the inter-season variability is higher at MSY, since we can appreciate a tendency towards higher C values during summer (JJA), as also observed for MSA. The likely reason for a higher inter-season variability at MSY and MSA compared to BCN was the fact that in BCN the SSA rarely reached values higher than around 0.95 for which a strong increase of the C was observed (cf. Fig. 1 in the manuscript). Moreover, as reported in the Fig. S7, the inter-season variability of the SSA mirrored quite well the inter-season variability of the C at the three stations.

We have included this figure reported below in the supplementary as Fig. S7.

[Figure]

**Figure S7.** Seasonal evolution of the SSA at a) BCN, b) MSY and c) MSA measurement stations for both M8020 and M8060 filter tapes. The box plot boxes show the range between the first and third quartile (IQR) with the median value for each season distribution represented by the inner line; the maximum whisker length is proportional to 1.5·IQR.

Also, we have modified the paragraph. Now it is in lines 394-408 as follows:

"As shown in Sect. 3.1, high SSA increased the C values, and, consequently, the C seasonality was affected, to some degree, by the SSA seasonality. In fact, Fig. S7 in the supplementary material shows that the seasonal evolution of the SSA at MSY and MSA mirrored quite well the seasonal evolution of the C, with an increase of both C and SSA toward the warm season. In BCN, the inter-season variability of both C and SSA was less pronounced and the C remained fairly constant during the different seasons. Exception was in the winter period (DJF) when both C (M8060) and SSA showed minima. Nevertheless, the variability within each season was the largest in BCN, due to a higher variability of the SSA values at this station within each season compared to MSY and MSA (Fig. S7). The relationship between C and SSA can be also observed in Fig. S8, where the diel cycles of both C and SSA were reported. In BCN, both C and SSA showed two relative minima in the morning and in the afternoon, mirroring the traffic rush hours. At MSY, the sea-breeze-driven transport of pollutants in the afternoon caused a reduction of both SSA and C. Conversely, at MSA both C and SSA showed less variability in the diel cycles and less similarity was observed. Note that the similarities commented above between the diel/seasonal cycles of C and SSA were more or less evident depending on the season/station considered. In fact, we have shown in Fig. 1 that high SSA (> 0.90-0.95) can strongly affect the C values, but less dependency between C and SSA was observed for lower SSA thus also contributing to mask the similarities between C and SSA reported in Figs. 3, S7, and S8 which were obtained averaging all available data, including C values at lower SSA.

The diel cycle of C and SSA, Fig. S7 in the supplementary, is:

BCN:

[Figure]

MSY:

[Figure]

MSA:

[Figure]

**Figure S8.** Diel evolution of the SSA and the C actor at a) BCN, b) MSY and c) MSA measurement stations.

**22. and seasonal cycles: Does the seasonality of ABL influence at MSA also influence the C seasonality? How do you explain that C from M8060 is larger in JJA at MSA but not C from M8020 ? How do you explain that the variability of C at BCN is smaller for M8020 than for M8060 and that the inverse situation (larger for M8020 than for M8060) at MSY? Do you find a diurnal variability of C that could be bounded to traffic intensity at BCN or to ABL influence at MSA?**

Indeed, seasonality and diel variations of the ABL at MSA have the potential to influence particle properties and sources observed at this site (e.g. Ripoll et al., 2014; Pandolfi et al., 2014). For example, Pandolfi et al. (2014) have shown that at MSA both scattering and absorption increase in the afternoon due to the transport of pollutants favored by thermally driven upslope winds and PBL height variations. Similarly, relatively lower scattering and absorption are usually measured at MSA in winter compared to other seasons/scenarios, because of the position of the MSA station often above the ABL. To a lesser extent, the same is typically observed at MSY. However, despite changes in absorption and scattering, the SSA at MSA do not change too much as it does at MSY station. Thus, the diel and seasonal cycles of SSA at MSA are usually less pronounced compared to MSY. This is the likely reason why we do not clearly see that the ABL strongly affected the analysis reported here. Moreover, as commented in the reply to the reviewer comment #21 above, the similarities commented above between the diel/seasonal cycles of C and SSA were more or less evident depending on the season/station. Moreover, we have shown in Fig. 1 that high SSA values (> 0.90-0.95) can clearly affect the C values, but almost no relationship between C and SSA was observed for lower SSA values thus also contributing to mask the similarities between the two variables reported in Figs. 3, S7, and S8.

Despite this, the differences in the seasonality of C can be partly explained with the seasonality of the SSA. As shown in the figure from the previous reviewer question (now Fig. S8), during the measurements with the M8020 at MSA in summer (JJA), the SSA was not as high as during the measurements with the M8060 filter tape for summer. Thus, the effect of SSA resulted in a lower C for the former filter. Unfortunately, we don't have the SSA data available for the M8020 period at BCN. We could only hypothesize that this difference in the variability between BCN and MSY station was due to a possible smaller SSA variability during the M8020 period at BCN than for the M8060 period.

Indeed, some exceptions are observed in the seasonal trend at MSA, such as the JJA period when the C for the M8020 filter is not higher compared to the other seasons. However, the C values during JJA are lower than those of the M8060 period, hence confirming that the M8020 filter has on average a lower fit. It is the MAM period which presents anomalous high values for the M8020 filter, which could be due to specific atmospheric conditions (a larger amount of Saharan dust intrusions) during the M8020 sampling period.

As commented in the previous reviewer comment #21, in BCN, both C and SSA showed two relative minima in the morning and in the afternoon, mirroring the traffic rush hours. At MSY, the sea-breeze-driven transport of pollutants in the afternoon caused a reduction of both SSA and C. Conversely, at MSA both C and SSA showed less variability in the diel cycles and less similarity was observed.

**23. and seasonal cycles (bis): You explain that secondary particle formation and Saharan dust outbreaks increase the variability of C. I think that a specific analysis of time with secondary particles/dust (and perhaps other parameters) would be very interesting and could help other stations with similar influences to determine their C values.**

Indeed, this specific analysis would be interesting. Although the proposed study could be of interest for further study, it falls out of the scope of this manuscript, and that previous studies of our group already describe the seasonal cycles of secondary particles and Saharan dust intrusions over the area of study, such as Ripoll et al. (2014)a,b at MSA, Ripoll et al. (2015) at MSY and MSA, and Querol et al. (2001) and Mohr et al. (2015) at BCN . What we can conclude is that, based on the demonstrated cross-sensitivity to scattering, the SSA is an important parameter affecting the C. Previous works have demonstrated that (e.g. Pandolfi et al., 2014) higher SSA at our measuring stations (at least MSY and MSA) are typically associated to Saharan dust outbreaks and summer regional recirculation episodes when, in this latter case, both secondary OA and sulfate concentrations increase. However, we recall (see also reply to the reviewer comment #21 above) that variations of SSA, for example with seasons/scenarios, are typically more pronounced at MSY rather than at MSA.

**24. Line 338: you cannot derive SSA value from the solely AE33.**

Indeed. We omitted the necessary scattering coefficient for obtaining the SSA. We have added it to the manuscript at lines 419-422:

"The observed increase of the C factor with wavelength affects the absorption coefficients derived from the AE33 attenuation measurements and, consequently, can affect all the intensive optical parameters such as AAE, SSA and SSAAE which can be derived from the multi-wavelength AE33 absorption measurements and scattering coefficient measurements.

**25. Figure 3: these results correspond to which tape? Why do we have Ceff comprised between 2-2.8 in Fig. 2 for all the stations and between 3-3.5 for BCN, 2.5-3 for MSY and 3.2-4.5 for MSA? The comparison with PP_uniMI is done on another set of data, but this large difference should be explained.**

The results obtained during the intercomparison with the PP_UniMI correspond to the currently used M8060 filter tape. The difference in the values of C are due to the offset of the absorption measurements between the MAAP (reference absorption in Fig. 2) and the PP_UniMI (reference absorption in Fig. 3). This offset in the absorption represents around a 20%, as can be seen in Fig. A1, and in Fig. 2 of Valentini et al. (2020).

We have included a paragraph mentioning this in lines 429-432:

"As can be appreciated by comparing Figs 2, 3, and 4, the multiple scattering correction factors obtained using the PP_UniMI reference instrument were larger than those obtained with the MAAP as a consequence of the offset in the absorption measurements between MAAP and PP_UniMI. A detailed discussion of this offset can be found in Fig. A1 and in Fig. 2 in Valentini et al. (2020)

**26. Line 340: If I remember well, Weingartner had not really the possibility to check the wavelength dependency of C with a reference.**

Indeed, Weingartner et al. (2013) had not the possibility to intercompare with a reference measurement. We have modified this sentence as follows in lines 424-428:

"For older versions of aethalometer (model AE31), Weingartner et al. (2003) found strong indication of the independence of C with the wavelength, and Segura et al. (2014) did not found any wavelength dependence of the multiple scattering parameter C with the wavelength. Conversely, Bernardoni et al. (2020) found a decrease of the C factor with wavelengths, although it was not statistically significant, and reported the impact of the wavelength dependent C on source apportionment model results.

Contradictory results have been reported in literature about the spectral dependence of C for older versionof aethalometer (model AE31). For example, Weingartner et al. (2003) found strong indication of the independence of C with wavelength, and neither Segura et al. (2014) found any wavelength dependence of the multiple scattering parameter C with the wavelength. Conversely, Bernardoni et al. (2020) observed a decrease of the C factor with wavelengths, although it was not statistically significant."

**27. Line 344-348: what is the relation with the C wavelength dependence since the loading correction is already applied in final AE33 data?**

We presented the work from Virkkula et al. (2015) and Drinovec et al. (2017) because we followed the same method to analysis the C wavelength dependence and its possible dependence with particles properties.

We have rearranged these paragraphs as follows (lines 459-463):

"To further explore the possible causes that contributed to the different C spectral dependencies observed, we performed a similar analysis as in Virkkula et al. (2015) by comparing the C and its wavelength dependence with different aerosol particles intensive optical properties, namely: SSA, BF and SSAAE. Virkkula et al. (2015) and Drinovec et al. (2017) have shown that the AE33 factor loading parameter, k, increases with increasing BF (smaller particles) and decreases with increasing SSA and that the wavelength dependence of k also depends on these two optical properties as well as on the particle mixing state."

**28. Line 351: Is it really necessary to introduce the abbreviations ac and ak since they are used only in this part of the study?**

We have modified this section. Now there is no reference to these abbreviations in the manuscript. As a consequence, aC is only used in Fig. S9 in the supplementary material, with the corresponding explanation in the caption.

**29. Figure 4 for MSA: there is no continuous increase of ac with SSA, but a shift at about SSA=0.95. Between SSA=0.85 and 0.94, there is even a very small decrease. Similarly there is no decrease with increasing BF since the second point (BF=0.1) is the lowest one. Regarding SSAAE, the two first points are clearly higher. Please give a better description and argumentation and perhaps a value for the statistical significance.**

We have modified and improve the description of this figure, which is now done in Fig. S9. Also, Figure 5 and the new figure 6 have been improved as requested by the referee:

[revised manuscript text omitted]

**30. Figure 4 (bis): I do not understand why there is an ac value of ~1.75 in subplot i) and not in subplots c) and f). Did you use another dataset? Similarly, why the ac value of 0.5 appears only in h) and not in b) and e) ?**

This discrepancy is due to the method employed for binning the data. We have followed the Freedman-Diaconis rule which defines the bin width. Since each of the rows of now Fig. S9 corresponds to a different variable (SSA, BF and SSAAE) each bin contains different data points.

In order to clarify this, we added the following sentence in the caption of Fig. S9:

"The values of aC (y-axis) for a given station changed depending on the dependent variable (x-axis) considered due to the method employed for binning the data. Here we used the Freedman-Diaconis rule to define the bin width that can, consequently, include different data points depending on the variable considered."

**31. Line 358: one of the lowest ac value corresponds also to a negative SSAAE. Please comment.**

Indeed, this point is the lowest. However, it does not change the general trend observed. It has to be noted that for some bins the number of data points is very scarce, in the new Fig. S9, we have colored the points with red when the data points at each bin where greater than 2, but lower than 5.

As mentioned in comment #29, the text that describes this dependency is at lines 459-477:

"To further explore the possible causes that contributed to the different C spectral dependencies observed, we performed a similar analysis as in Virkkula et al. (2015) by comparing the C and its wavelength dependence with different aerosol particles intensive optical properties, namely: SSA, BF and SSAAE. Virkkula et al. (2015) and Drinovec et al. (2017) have shown that the AE33 factor loading parameter, k, increases with increasing BF (smaller particles) and decreases with increasing SSA and that the wavelength dependence of k also depends on these two optical properties as well as on the particle mixing state. In Fig. S9 we present a similar

analysis by studying the effects of these intensive optical properties on the multiple scattering parameter C instead of k. Fig. S9 shows the slope of C with the wavelength (i.e. the wavelength-dependence of C) with SSA, BF, and SSAAE at the three sites. As reported in Fig. S9, no clear relationship was observed between the C slope and the three intensive optical properties at both BCN and MSY. Moreover, the C slope at these two sites were close to zero for the considered intensive optical properties. The observed lack of C gradients was again likely due to the fact that at BCN and MSY the SSA did not exceed the threshold value, even when the SSAAE indicated the possible presence of Saharan dust intrusions at MSY (cf. Fig. S9h). However, Fig. S9c shows that at MSA there was a shift of the C slope toward large positive values when SSA was above 0.95. Below this SSA threshold value, the C slope was close to zero confirming the reduced C wavelength dependence for low SSA values at MSA. Moreover, when the SSAAE/BF at MSA (cf. Fig. S9i and S9f) decreased towards negative/low values (Saharan dust intrusions), the slope of the C increased, again confirming the potential of coarser Saharan dust to increase the SSA and, consequently, the C especially at the remote site. Note that, as already commented (cf. Fig. 6), the C slope kept high positive values at MSA also for the samples not dominated by dust (SSAAE>0), thus further indicating the predominance effect of SSA on the C wavelength dependence. Thus, the results presented in Fig. S9 confirmed the effects of SSA on the C presented in Fig. 5 and 6."

**32. Figure 4 and p. 14:** *similarly to the discussion on the seasonal cycle, I would suggest you to isolate Saharan dust outbreaks to compute the wavelength dependence of C for these peculiar events. It would be very useful for the reader to know e.g. that the value of the C wavelength dependence in presence of dust.* **OK this is done in Fig. 5 (see my comment on reorganization): The results proven in Figure 5 should be discussed only with Figure 5 and not hypothetically with Fig. 4. Please reformulate all the discussion since the conclusion is that other particle properties contribute to positive ac observed at mountain top.**

As per suggested by the reviewer, we have rearranged the order and modify the discussion so that is now clearer and it does not lead to confusion.

As commented in the minor comments #29 and #3, it can be found between lines 459-477:

[revised manuscript text omitted]

**34. Lines 376-378: already explained in the previous §.**

Indeed, we have removed these lines to avoid repetition throughout the manuscript.

**35. Lines 379-391: a clear description of Fig. 4s should first mention that ak decreases continuously up to SSA=0.9 and increases rapidly for SSA larger values @ BCN (and idem for the other stations). Then you can describe the shift between positive and negative ak. Please note and comment that ak is also positive for SSA> ~0.96 so that very bright aerosol have a similar k behavior as very dark ones.**

Thanks to the reviewer comments, the layout of the paper changed considerably in the revised version of the manuscript. Results now are more focused on the key results of this work. For this reason, and if the reviewer agrees, we removed from section 3.3 the analysis of the compensation parameter k and its slope versus SSA, BF and SSAAE. In fact, this additional analysis distracts the reader from the real focus of this manuscript.

**36. Line 383-384: what are the potential conclusions from this sentence "The relationship between k and SSA was similar to that between ak and SSA"?**

The sentence was meant to express that the factor loading, k, decreased with the SSA, i.e. the more dispersive the particles are the less they require to be corrected. In the same way, the slope of k with the wavelength decreased with SSA, i.e. the more dispersive the particles, the higher the correction is at the shorter wavelengths compared to the longer wavelengths.

Nevertheless, due to the restructuration of the manuscript, as mentioned in the minor comment #35, this paragraph and the figure it was referring to in the supplementary have been removed.

**37. Line 400: and for 470 nm ?**

Indeed, it lacks the change for 470 nm. It has been omitted because it does not present a statistical change since the wavelength dependence of C is to increase with the wavelength, without affecting that much the lower wavelengths. We have added the following to lines 497-500 to avoid confusion.

"However, Fig. S10 shows a statistically significant increase of the SSA at MSA station of around 1.3% at 660 nm and 2% at 950 nm when using C($\lambda$) instead of C(const). Conversely, as expected, no statistically significant change was appreciated at the lower wavelength, 470 nm."

**38. Line 413-414: I think that at this point the reader is well aware of both types of filters. You can avoid the description.**

Following the reviewer suggestion, we have avoided repeating the description of the filter tapes in the main text, and used the abbreviation for differentiating each different filter tape.

**39. Table 1: please specify if the C reported by other authors is computed similarly to your's and to which of your C it corresponds.**

The method other authors have employed to report C was either through an average value or a Deming regression fit; we therefore included this information in the corresponding column of Table 1.

We have included a comment of the employed method by the different author in the table caption, to make it clearer and avoid confusion.

"Different approaches, as aforementioned in Section 3.2, have been used to obtain the factor C. Since the literature values are obtained through either one of the methods, we include these vales in its corresponding column (C or $C_{\text{Deming}}$)."

**40. Figure 7: Is there a unit problem between figure 7 and the text ? the ms values are a factor 100 larger in the text.**

The figure values are the fitted values of $m_s$, whereas the text express the percentage values. We have modified the $m_s$ values in now Fig. 1 so that these values are now coherent with the text. The new Fig. 1 is reported below.

[Figure]

---

## Author Comment (AC3)

*Referee comment on "Determination of the multiple-scattering correction factor and its cross-sensitivity to scattering and wavelength dependence for different AE33 Aethalometer filter tapes: A multi-instrumental approach" by Jesús Yus-Díez et al., Atmos. Meas. Tech. Discuss., https://doi.org/10.5194/amt-2021-46-RC1, 2021*

**Answer from the authors to referee #1**

On behalf of all the authors of the manuscript, we would like to acknowledge the work done in the review as well as the suggestions and comments for improving the study.

Hereafter we will answer and resolve the comments. Any minor comment, typo or writing corrections will be directly corrected in the manuscript.

**General comments**

**Offline absorption coefficient measurements were done using the PP_UniMI techique. However, it is not clear if the samples could have been affected between sampling time and measurement in Milan. The authors should provide more details on how the samples were handled and how long it took from sampling to analysis. One of the issues is, for example, that brown carbon could have been modified on the filter, thus affecting absorption wavelength dependence.**

The measurements in Milan were done between May and July 2019 (cf. table below) whereas the MAAP measurements were performed between October 2018 and June 2019 in BCN, between June 2018 and December 2018 at MSY, and between June 2018 and November 2018 at MSA. Therefore, the time between MAAP sampling and PP_UniMI measurements is the time between the time each spot was measured at the station and the PP_UniMI measurements. The samples, once extracted from the MAAP filter roll were stored in a fridge. Each spot was separated and saved in a petri dish and sent to Milan, where measurements started as soon as the samples arrived.

| Station | Measurement period |
|---|---|
| *Montsec* | 22/05/2019 to 29/05/2019 |
| *Barcelona* | 20/06/2019 to 25/06/2019 |
| *Montseny* | 26/07/2019 to 09/07/2019 |

Indeed, brown carbon concentrations could have been modified on the filter during this process.

We have specified this issue in the manuscript in the lines 200-204:

"The time elapsed between the MAAP measurements and the MAAP spots analysis with the PP_UniMI in Milan varied between one year and one month. Once selected and cut, each MAAP spot was stored in a petri dish in a fridge and then sent to Milan. We assumed that there were no major particle losses affecting the measured optical properties, although some volatile compounds could have been evaporated over the period.

**Section 3.1: How valid is a comparison between C values for the different tapes when the measurements were done at different times, with different aerosol**

**conditions/properties?**

Given the length of the measurement periods (Fig. S1), the measurements performed at the three stations covered a wide range of aerosol particles properties, thus we expect that the measurements well represented the aerosol conditions typically observed at the measurement stations. Therefore, we assume that the reported C values represented well the different filter tapes characteristics with minimal influence from different aerosol conditions.

We have added a sentence at the end of the last paragraph of Section 2.3.1 (lines 286-289) to point this out:

"Given the length of the measurement periods, we assumed that the AE33 filter tapes considered here were characterized under a wide range of aerosol particle properties typically observed at the measurement stations and that the non-simultaneity of AE33 measurements with the two filter tapes did not prevent the comparison between the obtained C.

**I understand the authors use a 3- or 7-wavelength log-log fit to retrieve AAE from Aethalometer measurements. However, this method is inaccurate. Please check https://doi.org/10.1140/epjb/e2004-00316-5**

We have, indeed, used a 7- wavelength log-log fit to retrieve AAE from the AE33. As commented by the referee, and as shown in Goldstein et al. (2004), this method of fitting the data introduces a greater error to the fit result. However, the fitted value remains fairly similar (Table 1 in Goldstein et al., 2004). Thus, since it is common practice in the aerosol community to derive the Ångström exponents through the log-log fit (e.g. Bergstrom et al., 2007; Ealo et al., 2016, Bernardoni et al., 2017), we prefer considering this method as a valid option for our work.

**One of the key arguments of the article is the wavelength dependence of the multiple scattering correction factor, C, at a remote station. Is this finding specific to remote stations? To remote stations subject to Saharan dust influence? Or only to this particular station? Please comment. It would be useful if the authors can provide other references showing similar findings.**

We acknowledge the reviewer for its comment and interest on the application to other stations. We have modified the structure of the manuscript and the approach to the analysis. We have specified its limitations and the possible extrapolations to other measurement stations. The main driver of the changes in the C value is its cross-sensitivity to scattering, therefore, remote stations with high SSA values, such as MSA, not only those affected by Saharan dust intrusions, may find of interest the results. Also, other regional background stations, when SSA is high, such as MSY, may find that the cross-sensitivity to scattering is having an impact on their measurements.

As far we know, no other similar study has been previously carried out that can show similar results; but we encourage the community to do so if they have the ability to carry out such intercomparison.

We have included a few remarks at the conclusion of the manuscript at lines 564-582 with the aim of providing a clearer guide:

"In summary, based on the results herein presented, the absorption coefficients from AE33 data can be corrected with different degrees of confidence depending on the information available to estimate the multiple scattering parameter C:

- A tailored dynamic multiple scattering parameter can be obtained if on-line simultaneous reference absorption measurements are available. In this case, a dynamic C with high

temporal resolution can be obtained, allowing an in-situ correction of AE33 data and allowing studying for example diel/seasonal cycles of the multiple scattering parameter. Here we used on-line MAAP absorption measurements at one wavelength for the determination of a dynamic C at the same MAAP wavelength.

- If independent reference multi-wavelengths absorption measurements are available, then the dependence of the multiple scattering parameter with wavelengths can be studied. Here we determined the wavelength dependence of the multiple scattering parameter by using the polar photometer (PP_UniMI) off-line absorption measurements performed on the MAAP filter spots and by comparing the off-line PP_UniMI measurements with AE33 attenuation data integrated over the MAAP filter spots time stamp.

- If reference absorption measurements are not available for the experimental determination of the C, then the average values of the multiple scattering parameter provided here for three different measurement stations can be used as reference.

- If both independent reference absorption measurements and scattering measurements are available, then the cross sensitivity to scattering of AE33 data can be determined by studying the relationship between C and single scattering albedo (SSA). In this case, a parameterization can be obtained relating C and SSA.

- If SSA measurements are not available, this work provides parameterized formulas that allow calculating C over a wide range of SSA values."

**Specific comments**

**L172: Please remove comma after "Thermo".**

Done.

**L174: Please avoid starting a sentence with an acronym.**

The sentence now starts as: "Black carbon, eBC, […]".

**L188: When were the PP_UniMI measurements done?**

As stated in the previous table, PP_UniMI measurements were performed between May and July of 2019.

| *Station* | **Measurement period** |
|---|---|
| *Montsec* | 22/05/2019 to 29/05/2019 |
| *Barcelona* | 20/06/2019 to 25/06/2019 |
| *Montseny* | 26/07/2019 to 09/07/2019 |

We have included this information in the lines 200-204 of the manuscript:

"The time elapsed between the MAAP measurements and the MAAP spots analysis with the PP_UniMI in Milan varied between one year and one month. Once selected and cut, each MAAP spot was stored in a petri dish in a fridge and then sent to Milan. We assumed that there were no major particle losses affecting the measured optical properties, although some volatile compounds could have been evaporated over the period.

**L214: Please remove comma after "Pty".**

Done.

**L275: Please detail how you calculated the AAE.**

The AAE was calculated through a log-log linear fit to the 7 absorption wavelengths measured by the AE33. We include a clarification of this in the manuscript lines 294-296, in Sect. 2.3.2:

"The absorption coefficients from the PP_UniMI were inter/extrapolated to the seven AE33 wavelengths using the attenuation Ångström exponent, obtained through a log-log from the PP_UniMI absorption measurements."

**Fig 2: You mentioned previously that high SSA was observed in summer season and it increases with C but here it is shown that the highest C values are reached in winter, at least for MSY and MSA.**

We can see in Fig. 2 that on average the annual cycle of the C at MSY and MSA showed an increase during the summer period, corresponding with an increase of the SSA (new Fig. S7). At MSY, the annual cycle of the M8020 C was less pronounced but it still mirrored the variability of the SSA observed during the period with the M8020 filter tape. (Fig. S7**.)**

Here we present the now Fig. S7, which shows the SSA seasonal analysis.

[Figure]

**Figure S7.** Seasonal evolution of the SSA at a) BCN, b) MSY and c) MSA measurement stations for both M8020 and M8060 filter tapes. The box plot boxes show the range between the first and third quartile (IQR) with the median value for each season distribution represented by the inner line; the maximum whisker length is proportional to 1.5·IQR.

**L370: I guess the "3.3" is a typo.**

Corrected the typo.

**L423: Could you please provide more references here? Other remote sites with SSA > 0.95?**

We have increased the references at the line to:

"(Collaud Coen et al., 2004; **Gyawali et al., 2009**; **Andrews et al., 2011**; Pandolfi et al., 2014a, 2018; **Schmeisser et al., 2018**; **Ferrero et al., 2019**; **Laj et al., 2020**)"

With the new references being:

1       Gyawali, M., Arnott, W. P., Lewis, K. & Moosmüller, H. In situ aerosol optics in Reno, NV, USA during and after the summer 2008 California wildfires and the influence of absorbing and non-absorbing organic coatings on spectral light absorption. *Atmos. Chem. Phys.* **9**, 8007–8015 (2009).
2       Andrews, E. *et al.* Climatology of aerosol radiative properties in the free troposphere. *Atmos. Res.* **102**, 365–393 (2011).
3       Schmeisser, L. *et al.* Seasonality of aerosol optical properties in the Arctic. *Atmos. Chem. Phys.* **18**, 11599–11622 (2018).
4       Ferrero, L. *et al.* Aerosol optical properties in the Arctic: The role of aerosol chemistry and dust composition in a closure experiment between Lidar and tethered balloon vertical profiles. *Sci. Total Environ.* **686**, 452–467 (2019).
5       Laj, P. *et al.* A global analysis of climate-relevant aerosol properties retrieved from the network of GAW near-surface observatories. *Atmos. Meas. Tech.* **13**, 4353–4392 (2020).

**Bibliography.**

1.      Goldstein, M. L., Morris, S. A. & Yen, G. G. Problems with fitting to the power-law distribution. *Eur. Phys. J. B* **41**, 255–258 (2004).
2.      Bergstrom, R. W. *et al.* Spectral absorption properties of atmospheric aerosols. *Atmos. Chem. Phys.* **7**, 5937–5943 (2007).
3.      Ealo, M. *et al.* Detection of Saharan dust and biomass burning events using near-real-time intensive aerosol optical properties in the north-western Mediterranean. *Atmos. Chem. Phys.* **16**, 12567–12586 (2016).
4.      Bernardoni, V., Valli, G. & Vecchi, R. Set-up of a multi wavelength polar photometer for off-line absorption coefficient measurements on 1-h resolved aerosol samples. *J. Aerosol Sci.* **107**, 84–93 (2017).

---

## Author Comment (AC5)

[revised manuscript text omitted]
570 with the observed lack of dependence of the C factor with wavelength. However, Fig. S10 shows a statistically significant increase of the SSA at MSA station of around 1.3%  at 660 nm  and 2%  at 950 nm compared to a constant C. when using C(λ) instead of C(const). Conversely, as expected, no statistically significant change was appreciated at the lower wavelength, 470 nm. This variation introduced by C(λ) on AAE and SSA, although not large, is relevant since it occurs at the threshold of SSA value for which a substantial increase of the C
575 as a function of SSA was observed, as shown in Section 3.1.

**3.4**

580 ## 4 Conclusions

Here, the multiple scattering parameter C for two filter tapes used in AE33 dual-spot aethalometers, i.e. the previously used M8020 and the currently used M8060 filter tapes, has been analyzed using data collected at three different background stations in NE Spain: an urban background station in Barcelona, BCN, a regional background station at Montseny, MSY, and a mountain-top station at Montsec d'Ares, MSA. We obtained the C correction factor comparing the AE33 attenuation
585 measurements with the absorption coefficients measured from MAAP instruments, and used simultaneous scattering measurements from an integrating nephelometer to characterize the cross-sensitivity to scattering of  C. Moreover, we studied the C wavelength dependence at the three

590  sites comparing the  measurements with the multi-wavelength PP_UniMI absorption coefficients.

 We presented here a novel approach to characterize the cross-sensitivity
595 to scattering of the C correction factor. This approach consisted in fitting the measurements of the C versus SSA. The fits provided the constant $C_f$ and a cross-sensitivity factor $m_S$. We have applied the fits to the M8020 filter tape at MSY and MSA  and we obtained higher cross-sensitivity values of the C factor ($1.8\pm0.1\%$ and $3.4\pm0.1\%$, respectively) compared to those reported in the literature (around 1-1.5 %). For the  first time
600 here we characterized the cross-sensitivity to scattering also of the new M8060 filter tape. We obtained a cross-sensitivity to scattering  for the M8060 of $1.6\pm0.3\%$, $3.0\pm0.1\%$ and $4.9\pm0.1\%$ for BCN, MSY and MSA, respectively. The multiple scattering parameter, $C_f$, for the M8020 filter tape was $2.21 \pm 0.01$ at MSY and $1.96 \pm 0.02$ at MSA. For the M8060 filter tape  the fit led to $C_f$ values of $2.50 \pm 0.02$ at BCN,
605  $1.96 \pm 0.01$ at MSY, and  $1.82 \pm 0.02$ at MSA.

610

2.44 ± 0.57 2.20 ± 0.02 2.50 ± 0.02 1.6 ± 0.3Leipzig Urban background TFE ? 3.2 TFE ? 2.78 Rome Urban background M8060 ? 2.66 Klagenfurt Urban background TFE ? 1.57 **Montseny** Regional background TFE 
[revised manuscript text omitted]

[Figure]

**Figure S1.** Multiple scattering parameter (C) availability for both M8060 and TFE filter tape at BCN, MSY and MSA measurement supersites.

[Figure]

**Figure S2.** Normalized count distribution of the measurement timestamp, $\delta t$ in minutes for a) BCN, b) MSY, and c) MSA. Time measurement resolution was set to 1 min when possible, in b) and c) the 5 min spikes are due to a measurement time resolution of 5 min during a certain period of time.

| INSTRUMENT | STATION | TIMESTAMP |
|---|---|---|
| **AE33** | BCN | 1 min |
| | MSY | 1 min |
| | MSA | 1 min |
| **MAAP** | BCN | 1 min |
| | MSY | 1 min |
| | MSA | 1 min |
| **NEPHELOMETER** | BCN | 1 min |
| | MSY | 5 min (2013-February February 2017); 1 min (February 2017-2020) |
| | MSA | 5 min (2013-February February 2017); 1 min (February 2017-2020) |

**Table S2.** Timestamp of the measurement for each instrument, AE33, MAAP and nephelometer, for each station.

[Figure]

**Figure S3.** Multiple scattering parameter (C) dependence on the single scattering albedo (SSA) for the TFE-coated glass (upper panel) and the M8060 filter tape (lower panel) at: BCN (c), MSY (a,d) and MSA (b,e) measurement supersites as a function of the absorption Ångström exponent (AAE).

[Figure]

**Figure S4.** Multiple scattering parameter (C) dependence on the single scattering albedo (SSA) for the TFE-coated glass (upper panel) and the M8060 filter tape (lower panel) at: BCN (c), MSY (a,d) and MSA (b,e) measurement supersites as a function of the backscattered fraction at (BF).

[Figure]

**Figure S5.** Multiple scattering parameter (C) dependence on the single scattering albedo (SSA) for the TFE-coated glass (upper panel) and the M8060 filter tape (lower panel) at: BCN (c), MSY (a,d) and MSA (b,e) measurement supersites as a function of the single-scattering albedo Ångström exponent (SSAAE).

[Figure]

**Figure S6.** Scatter-plot of the binned AE33 attenuation coefficient ($b_{atn}$ (637 nm)) vs MAAP absorption coefficient $b_{abs}$ (637 nm)) where the slope of the Deming regression, m, represents the multiple-scattering parameter C, and q is the intercept of the regression, for the TFE-coated glass filter tape (upper panels) and M8060 filter tape (lower panels) for BCN (a,d), MSY (b,e) and MSA (c,f). The non-zero intercept, q, is indicative of the additional signal due to the cross-sensitivity to scattering of particles within the filter.

[Figure]

**Figure S7.** Seasonal evolution of the SSA at a) BCN, b) MSY and c) MSA measurement stations for both M8020 and M8060 filter tapes. The box plot boxes show the range between the first and third quartile (IQR) with the median value for each season distribution represented by the inner line; the maximum whisker length is proportional to 1.5·IQR.

a)

[Figure]

b):

[Figure]

c):

[Figure]

**Figure S8.** Diel evolution of the SSA and the C actor at a) BCN, b) MSY and c) MSA measurement stations.

|  | $C_{PP\_UniMI}(\lambda)$ | | | | | | |
|---|---|---|---|---|---|---|---|
|  | **370 nm** | **470 nm** | **520 nm** | **590 nm** | **660 nm** | **880 nm** | **950 nm** |
| *BCN* | 3.36 | 3.26 | 3.22 | 3.24 | 3.21 | 3.19 | 3.31 |
| *MSY* | 2.68 | 2.67 | 2.72 | 2.77 | 2.79 | 2.62 | 6.67 |
| *MSA* | 3.47 | 3.48 | 3.58 | 3.71 | 3.87 | 4.05 | 4.03 |

**Table S2.** Multiple scattering factor (C) at each AE33 measuring wavelength obtained using the absorption coefficient from the PP_UniMI polar photometer for BCN, MSY and MSA measurement supersites.

| | $C_{PaM}(\lambda)$ | | | | | | |
|---|---|---|---|---|---|---|---|
| | **370 nm** | **470 nm** | **520 nm** | **590 nm** | **660 nm** | **880 nm** | **950 nm** |
| BCN | 2.82 | 2.78 | 2.75 | 2.73 | 2.72 | 2.69 | 2.83 |
| MSY | 2.32 | 2.33 | 2.42 | 2.46 | 2.47 | 2.26 | 2.32 |
| MSA | 2.82 | 2.85 | 2.91 | 3.03 | 3.09 | 3.22 | 3.24 |

**Table S3.** Multiple scattering factor (C) at each AE33 measuring wavelength obtained using the absorption coefficient from the PP_UniMI polar photometer working as MAAP (PaM) for BCN, MSY and MSA measurement supersites.

[Figure]

**Figure S9.** Relationship between the slope of the factor C and the wavelength, $a_C$, and the single-scattering albedo at 520 nm ($SSA_{520nm}$), the backscatter fraction ($BF_{520nm}$), and the single-scattering albedo Ångström exponent (SSAAE) at BCN (left panel), MSY (middle panel) and MSA (right panel) measurement stations. The values of $a_C$ (y-axis) for a given station changed depending on the dependent variable (x-axis) considered due to the method employed for binning the data. Here we used the Freedman-Diaconis rule to define the bin width that can, consequently, include different data points depending on the variable considered. The red points show bins with a number of measurements which range between 2 and 5 data points.

|       | AAE            |                |
|-------|----------------|----------------|
|       | $C(const)$     | $C(\lambda)$   |
| **BCN** | $1.19 \pm 0.15$ | $1.17 \pm 0.15$ |
| **MSY** | $1.27 \pm 0.12$ | $1.25 \pm 0.12$ |
| **MSA** | $1.19 \pm 0.07$ | $1.35 \pm 0.07$ |

**Table S4.** Mean values of the absorption Ångström exponent (AAE) for the sensitivity analysis performed in Fig. 3 on the AAE obtained using a wavelength-dependent C ($C(\lambda)$) in comparison with an AAE obtained using a constant C, C(const), parameter.

a)

[Figure]

b)

c)

[Figure]

**Figure S10.** Sensitivity analysis of the single scattering albedo (SSA) on the wavelength-dependent C (C(λ)) in comparison with an SSA at 3 wavelengths (470, 660 and 950 nm) obtained using a constant C parameter (C(const)) for a) BCN, b) MSY and c) MSA measurement stations.